# The effect of secondary ice production parameterization on the simulation of a cold frontal rainband

Sylvia C. Sullivan[1,2], Christian Barthlott[1], Jonathan Crosier[3], Ilya Zhukov[4], Athanasios Nenes[2,5,6], and Corinna Hoose[1]

[1]Institute of Meteorology and Climate Research, Karlsruhe Institute of Technology, Karlsruhe, Germany
[2]Department of Chemical and Biomolecular Engineering, Georgia Institute of Technology, Atlanta, GA 30332, USA
[3]School of Earth, Atmospheric, and Environmental Studies, University of Manchester, Manchester, UK
[4]Jülich Supercomputing Center, Forschungszentrum Jülich, Jülich, Germany
[5]ICE-HT, Foundation for Research and Technology, Hellas, 26504 Patras, Greece
[6]Institute of Environmental Research and Sustainable Development, National Observatory of Athens, 15236, Palea Penteli, Greece

*Correspondence to:* S. Sullivan (scs2229@columbia.edu), A. Nenes (athanasios.nenes@gatech.edu), C. Hoose (corinna.hoose@kit.edu)

**Abstract.** Secondary ice production via processes like rime splintering, frozen droplet shattering, and breakup upon ice hydrometeor collision have been proposed to explain discrepancies between in-cloud ice crystal and ice-nucleating particle numbers. To understand the impact of this additional ice crystal generation on surface precipitation, we present one of the first studies to implement frozen droplet shattering and ice-ice collisional breakup parameterizations in a mesoscale model. We simulate a cold frontal rainband from the Aerosol Properties, PRocesses, And InfluenceS on the Earth's Climate campaign and investigate the impact of the new parameterizations on the simulated ice crystal number concentrations (ICNC) and precipitation. Near the convective regions of the rainband, contributions to ICNC can be as large from secondary production as from primary nucleation, but ICNCs greater than $50 \, \mathrm{L}^{-1}$ remain underestimated by the model. Addition of the secondary production parameterizations also clearly intensifies the differences in both accumulated precipitation and precipitation rate between the convective towers and non-convective gap regions. We suggest, then, that secondary ice production parameterizations be included in large-scale models on the basis of large hydrometeor concentration and convective activity criteria.

## 1 Introduction

Cloud microphysics mediate precipitation formation, either from the in-cloud liquid or ice phase. In both cases, precipitation is observed to form much faster than the time frame to form sedimentable hydrometeors by solely condensational or depositional growth. Instead, accretional growth is required, be it collision-coalescence of liquid droplets, droplet riming on ice hydrometeors, or ice crystal aggregation. The efficiency of these processes is controlled by hydrometeor size through their terminal velocity and cross section within a collisional kernel (e.g., Rosenfeld and Gutman, 2001; Khain et al., 2005).

   In clouds with high cloud condensation nuclei (CCN) concentrations, precipitation is more likely to initiate in the ice phase because cloud droplets will be smaller and less likely to grow to sedimentable size. This ice-initiated precipitation occurs often

over the continents, where aerosol loadings are higher (e.g., Mülmenstädt et al., 2015; Lohmann, 2017), and in convective clouds for which the vertical motions are strong enough to carry droplets above the freezing level. Cold phase initiation has been associated with the top 10% of heavier rains according to data from the Tropical Rainfall Measuring Mission (Lau and Wu, 2011), and precipitation indices have been developed based upon cold cloud coverage (e.g. Arkin and Meisner, 1987; Joyce and Arkin, 1997). In these cases of ice-initiated precipitation, the requisite crystal growth can occur via riming or the Bergeron process, in which water vapor transfers from droplets to ice crystals under appropriate thermodynamic conditions. Ice hydrometeors eventually fall out of the cloud and reach an altitude at which they melt to form rain drops.

To accurately forecast cold phase-initiated precipitation, we must first accurately model in-cloud ice formation. Much effort has been devoted to understanding which atmospheric aerosols can act as ice-nucleating particles (INP) (e.g., Möhler et al., 2006; Knopf and Koop, 2006; Möhler et al., 2008; DeMott et al., 2010; Broadley et al., 2012; Hoose and Möhler, 2012; O'Sullivan et al., 2015; DeMott et al., 2016). But numerous measurements also indicate much larger in-cloud ice crystal number concentrations (ICNC) than INP numbers (e.g., Crawford et al., 2012; Heymsfield and Willis, 2014; Lasher-Trapp et al., 2016; Taylor et al., 2016). These kinds of ICNC 'enhancements' may be due to shattering upon impact with the probe inlet (e.g., Heymsfield, 2007; McFarquhar et al., 2007), but more recent measurements employ probe tips that reduce airflow disturbance around the inlet (Korolev et al., 2013a, b) and interarrival time (IAT) algorithms to filter out artifacts (Field et al., 2003, 2006; Korolev and Field, 2015). The ICNC-INP discrepancy remains in some cases, and a variety of secondary ice production processes have been proposed to make up the difference.

Given the linkage of cloud ice and precipitation, including these secondary ice production processes in meteorological models may yield more accurate precipitation forecasts. But the precipitation change with ICNC change will not always be the same, as schematized in Figure 1. For example, an additional source of small ice crystals would extend cloud duration and delay precipitation in a kind of cloud lifetime effect. But the depositional growth of these small ice crystals may also deplete supersaturation to a level at which the Bergeron process initiates. Then ice hydrometeors quickly become large, shortening cloud duration and accelerating precipitation. This pathway should be more important for stratiform precipitation, given the narrow range of requisite ambient vapor pressures: indeed for an integral ice radius of 100 $\mu$m cm$^{-3}$, the updraft must be less than about a 1 m s$^{-1}$ for the Bergeron process to occur (Korolev, 2007). A 'cascade effect' has been proposed in which small ice crystals collide with large droplets, freezing and shattering them and forming more crystals (Lawson et al., 2015). Dynamic-microphysical feedbacks could exist: an additional source of small crystals would generate more latent heat, changing the vertical heating profile and potentially affecting precipitation through altered detrainment rates or cloud updrafts (e.g. Clark et al., 2005). Efficient riming at mixed-phase temperatures may also simply generate larger hydrometeors that sediment more quickly, particularly in convective regions with a high degree of mixing. Along with these hydrological implications of altered precipitation, more glaciated clouds will be optically thinner and radiatively warm the surface.

Several studies have considered these linkages, both with measurements and models, but no consensus has been reached on their individual or net impacts. For example, Connolly et al. (2006b) did not see a large change in surface precipitation from a tropical thunderstorm when they altered the rime splintering rate in the Weather Research and Forecasting model. Dearden et al. (2016) also found that, in simulations of summertime cyclones, depositional growth of ice crystals was much

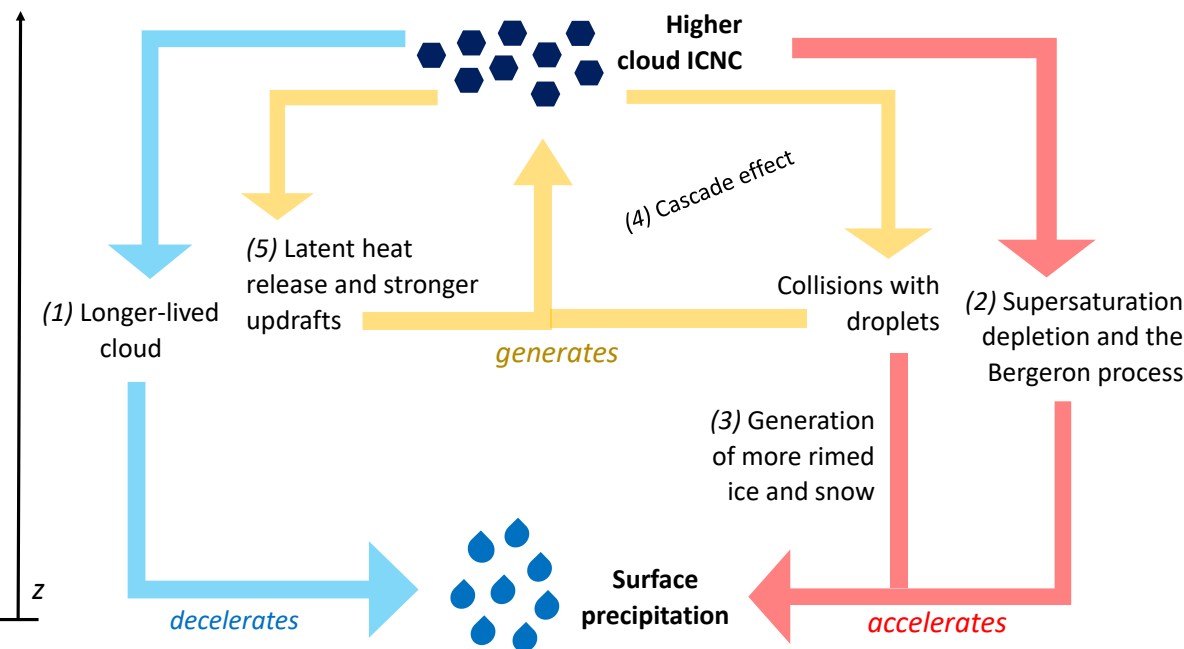

**Figure 1.** Within mixed-phase clouds, ice and its secondary production will impact surface precipitation in various ways. Precipitation rates could decelerate via a cloud lifetime effect (1), where small crystals stay aloft longer, or accelerate via supersaturation depletion (2), where larger crystals grow at the expense of liquid droplets via the Bergeron process. More frequent riming may simply generate more large and dense hydrometeors that can fall out and melt to generate more surface precipitation (3). Additional crystals due to droplet freezing could also generate a cascade effect (4) whereby the generation of more ice leads to more collisions with droplets that leads to generation of more ice. The latent heat release upon droplet freezing can also invigorate updrafts (5).

more influential than inclusion of rime splintering on the spatial distribution of precipitation. On the other hand, an early study by Aleksić (1989) found that, in Serbia, more numerous ice crystals due to a hail suppression program led to more intense rainfall. Clark et al. (2005) discussed how the latent heating from additional ice generation modifies the vertical temperature profile, and hence precipitation rates. And Taylor et al. (2016) concluded that the combination of droplet coalescence and secondary ice production often determined precipitation timing and intensity in the maritime cumuli they observed. The kind of compensating effects discussed above, along with the insusceptibility of accretional processes to aerosol perturbations, would also reduce sensitivity of precipitation to aerosol more generally, as discussed in the study of Glassmeier and Lohmann (2016).

Here, we implement parameterizations of frozen droplet shattering and ice-ice collisional breakup, two proposed secondary production processes, into the Consortium for Small-Scale modeling (COSMO) framework described in Section 3. Frozen

droplet shattering refers to a pressure build-up as a larger droplet freezes, either due to formation of an external ice shell or to latent heat of fusion release. This increased pressure eventually leads to spicule ejection or cracking and explosion of the ice shell (Leisner et al., 2014; Wildeman et al., 2017). In ice-ice collisional breakup, the impact of two ice hydrometeors leads to
shattering, particularly of dendrites or fragile protuberances (Vardiman, 1978; Takahashi et al., 1995; Yano and Phillips, 2011). COSMO already includes a description of a third process called rime splintering (Hallett and Mossop, 1974). We investigate the impact of these parameterizations on the simulated ICNC and surface precipitation in a case study.

## 2 Parameterizations

### 2.1 Frozen droplet shattering parameterization

Recent droplet levitation experiments and high speed video are elucidating the exact physics behind the shattering of droplets as they freeze (Leisner et al., 2014; Wildeman et al., 2017). Droplet shattering has been previously parameterized statistically in a bin microphysics scheme with the fragment number as a function of drop diameter to the fourth power, using data from the Ice in Clouds Experiment - Tropical (ICE-T) campaign (Lawson et al., 2015, 2017). But while measurements continue to confirm a strong dependence of fragment number on droplet size, even recent studies could not confirm this fourth-power
dependence (e.g. Lauber et al., 2018). The laboratory studies of Lauber et al. (2018) in particular add important quantitative results to existing secondary ice measurements but are taken at two droplet sizes (83 and 310 $\mu$m) so that it remains difficult to rigorously formulate fragment number. For now, we parameterize this process with the product of a fixed fragment number, a temperature-dependent shattering probability, and the existing droplet freezing tendency:

$$\left.\frac{\partial N_{ice}}{\partial t}\right|_{DS} = \left[1 + p_{DS}\aleph_{DS}\right]\frac{\partial N_{freez}}{\partial t} \tag{1}$$

$$p_{DS} = \frac{p_{max}}{\max\{\mathcal{N}(T_\mu,\sigma)\}}\mathcal{N}(T_\mu,\sigma) \tag{2}$$

$$\mathcal{N}(T_\mu,\sigma) = \frac{1}{\sqrt{2\pi}\sigma}\exp\left[\frac{-(T-T_\mu)^2}{2\sigma^2}\right] \tag{3}$$

A constant value for the ejected fragment number $\aleph_{DS}$ is used, and shattering probability is given by a normal distribution in temperature, centered at a temperature $T_\mu$ of 258 K, with a default standard deviation $\sigma$ of 3 K and default maximum probability $p_{max}$ is 20%, similar to Sullivan et al. (2018). Examples of this distribution are shown in Figure 2b and d. Then
$\partial N_{freez}/\partial t$ is the number of frozen raindrops for a given time step, predicted by the Bigg (1953) heterogeneous freezing parameterization (see Appendix A for details).

### 2.2 Ice-ice collisional breakup parameterization

Vardiman first parameterized ice-ice collisional breakup using fragment generation functions based on the momentum exchange between two particles upon impact and leading coefficients dependent upon crystal type (Vardiman, 1978). More

recently, Yano and Phillips (2011) and Yano et al. (2016) constructed a dynamical system-type models that tracks only ice crystal, small graupel, and large graupel number densities and illustrated the ability of ice-ice collisions to generate huge ice crystal enhancements in the absence of vapor limitation. Recently a more complete parameterization has used an exponential formulation with the initial kinetic energy of two particles, their temperature- and humidity-dependent collision type, and asperity fragility coefficients (Phillips et al., 2017b, a). We choose to focus on temperature dependence in a more straightforward, if less physically rigorous, product of fragment number and hydrometeor collision tendency:

$$\frac{\partial N_{ice}}{\partial t}\bigg|_{BR,jk} = -\aleph_{BR}\frac{\partial N_j}{\partial t}\bigg|_{coll,jk} \tag{4}$$

$$\aleph_{BR} = F_{BR}(T - T_{min})^{1.2}\ \exp\bigg[-(T - T_{min})/\gamma_{BR}\bigg] \tag{5}$$

$$\frac{\partial N_j}{\partial t}\bigg|_{coll,jk} = -\frac{\pi}{4}\overline{E_{jk}}N_j N_k\big[\delta_j^0 D_j^2(\overline{x_j}) + \delta_{jk}^0 D_j(\overline{x_j})D_k(\overline{x_k}) + \delta_k^0 D_k^2(\overline{x_k})\big]$$

$$\times \big[\theta_j^0 v_j^2(\overline{x_j}) - \theta_{jk}^0 v_j(\overline{x_j})v_k(\overline{x_k}) + \theta_k^0 v_k^2(\overline{x_k})\big]^{1/2} \tag{6}$$

where the number of fragments generated $\aleph_{BR}$ is based upon the laboratory data of Takahashi et al. (1995) as in Sullivan et al. (2017). Within Equation 5, $T_{min}$ is a minimum temperature below which no breakup occurs and $\gamma_{BR}$ is the decay rate of fragment number at warmer subzero temperatures. The effect of these parameters, along with the leading coefficient $F_{BR}$, is illustrated in Figure 2a and c. The collisional tendency $\partial N_j/\partial t_{coll,jk}$ involves a collection efficiency $E_{jk}$; the number, mass, and terminal velocities of the colliding hydrometeors; and nondimensional values $\delta^0$ and $\theta^0$ from a gamma size distribution (see Appendix B for details). The parameterization works with four ice hydrometeor classes, shown in Table 1, by designating one hydrometeor class as the 'collider' ($j$) and a second as the 'collided' ($k$). The number in the 'collided' class is increased by $\aleph_{BR}$, while that in the 'collider' class remains constant. Mass in both classes remains unchanged. Future studies could include collisions between all hydrometeor classes or redistribute number and mass between the 'collider' and 'collided' differently. These considerations could also be obviated by implementing the parameterization within a property-based ice microphysics scheme like the Predicted Particle Properties (P3) scheme (Morrison and Milbrandt, 2015; Milbrandt and Morrison, 2015). P3 tracks ice mixing ratio, number, mass, and rime fraction rather than number and mass in snow, graupel, and ice crystal categories whose thresholds can be non-physical. We expect a strong influence of temperature from our breakup tendency $(\partial N_{ice}/\partial t)_{BR,jk}$ than was discussed in Phillips et al. (2017a), given the direct and sole dependence in Equation 6.

## 2.3 Rime splintering parameterization

The existing parameterization of rime splintering is a product of a leading coefficient, a temperature-dependent weighting, and a rime mixing ratio:

$$\frac{\partial N_{ice}}{\partial t}\bigg|_{RS} = \aleph_{RS}w_{RS}q_{rim}. \tag{7}$$

**Table 1.** New subroutines in the two-moment scheme, containing the ice hydrometeor collision parameterizations, given as *collider_ breakup_ collided*.

|  | ice crystals | snow | graupel | hail |
|---|---|---|---|---|
| **ice crystals** |  |  | *graupel_ breakup_ ice* | *hail_ breakup_ ice* |
| **snow** |  |  | *graupel_ breakup_ snow* |  |
| **graupel** | *graupel_ breakup_ ice* | *graupel_ breakup_ snow* |  | *hail_ breakup_ graupel* |
| **hail** | *hail_ breakup_ ice* |  | *hail_ breakup_ graupel* |  |

A default value of 3.5 $\times 10^8$ fragments per kilogram of rime, $\aleph_{RS}$, is used base on the experiments of Hallett and Mossop (1974). $q_{rim}$ denotes the rime mixing ratio, and $w_{RS}$ denotes a temperature-dependent weighting for rime splintering, which is triangular (TR) between 265 and 270 K by default and shown in Figure 2a and b. We add a second, uniform temperature weighting (UNI) between 263 and 273 K to investigate the possibility of a droplet shattering or ice-ice collisional breakup 'trigger' that feeds into a rime splintering 'cascade'. The rimer surface temperature may in fact be the more important factor

and can remain between 265 and 270 K, even for cloud temperatures a few degrees colder (Heymsfield and Mossop, 1984). We also limit rime splintering to occur only after collisions between cloud droplets of diameter greater than 25 $\mu$m or raindrops and ice crystals, graupel, hail, or snow (e.g., Phillips et al.; Connolly et al., 2006a).

## 3   Simulations

These parameterizations and adjustments are implemented into the Consortium for Small-Scale modeling (COSMO) frame-

work version 5.03 (Baldauf et al., 2011; Doms and Baldauf, 2015) that employs the two-moment microphysics scheme of Seifert and Beheng (2006) (SB06 hereafter). Several sensitivity tests are run, as listed in Table 2 and visualized in Figure 2. Simulation names include **RS** for rime splintering, **DS** for droplet shattering, or **BR** for ice-ice collisional breakup. Two sets of parameters, one weaker (denoted **1**) and one stronger (denoted **2**), are defined for each tendency. We also run a set of combination simulations in which multiple or all of the secondary ice parameterizations are simultaneously activated (**RS+BR**,

**RS+DS**, **DS+BR**, and **ALL**) and a control simulation in which all secondary ice production processes, including the default rime splintering in SB06, are turned off (denoted **CTRL** throughout). For each simulation, an ensemble of 5 runs is done with "stochastically perturbed physics tendencies" (e.g., Buizza et al., 1999), applied to vapor, cloud, and ice mixing ratio tendencies, and the ensemble mean output and standard deviation are evaluated.

We use these parameterization configurations to simulate 3 March 2009 for a domain centered at 53°N, 5°W, with longitudes

ranging from 65°N down to 46°N and latitudes ranging from 18°W to 10°E. In-situ cloud ice data and remote-sensing rainfall data are available for this case from the Aerosol Properties, PRocesses And InfluenceS on the Earth's climate (APPRAISE) campaign and Chilbolton Facility for Atmospheric and Radio Research (CFARR) in Southern England respectively. The observations showed the passage of a narrow cloud frontal rain band over the UK around 18:00 UTC and CFARR at 20:00 UTC. These data have been thoroughly analyzed by Crosier et al. (2014).

**Table 2.** Sensitivity tests are listed with $\aleph_{RS}$ in units of $(\text{mg rime})^{-1}$ and $T_{min}, \sigma$ in Kelvin.

| Rime splintering | | | Ice-ice collisional breakup | | | |
|---|---|---|---|---|---|---|
| **RS1**: | $\aleph_{RS} = 300$, | $w_{RS} = \text{TR}$ | **BR1ig**: | *graupel_breakup_ice* | | |
| | | | | $F_{BR} = 180$, | $T_{min} = 256$, | $\gamma = 3$ |
| **RS2**: | $\aleph_{RS} = 300$, | $w_{RS} = \text{UNI}$ | **BR2ig**: | *graupel_breakup_ice* | | |
| | | | | $F_{BR} = 360$, | $T_{min} = 249$, | $\gamma = 5$ |
| | | | **BR2sg**: | *graupel_breakup_snow* | | |
| | | | | $F_{BR} = 360$, | $T_{min} = 249$, | $\gamma = 5$ |

| Droplet shattering | | | Combinations | | | |
|---|---|---|---|---|---|---|
| **DS1**: | $\aleph_{DS} = 2$, | $p_{max} = 5\%$, | $\sigma = 3$ | **RS2 + BR2ig**: | $\aleph_{RS} = 300$, | $w_{RS} = \text{UNI}$ |
| | | | | | *graupel_breakup_ice* | |
| | | | | | $F_{BR} = 360$, | $T_{min} = 249$, $\gamma = 5$ |
| **DS2**: | $\aleph_{DS} = 10$, | $p_{max} = 10\%$, | $\sigma = 5$ | **DS2 + BR2ig**: | $\aleph_{DS} = 10$, | $p_{max} = 10\%$, $\sigma = 5$ |
| | | | | | *graupel_breakup_ice* | |
| | | | | | $F_{BR} = 360$, | $T_{min} = 249$, $\gamma = 5$ |
| | | | | **RS2 + DS2**: | $\aleph_{RS} = 300$, | $w_{RS} = \text{UNI}$ |
| | | | | | $\aleph_{DS} = 10$, | $p_{max} = 10\%$, $\sigma = 5$ |

| Control | | | | | | |
|---|---|---|---|---|---|---|
| **CTRL**: | $\aleph_* = 0$ | $F_{BR} = 0$ | **ALL**: | $\aleph_{RS} = 300$, | $w_{RS} = \text{UNI}$ | |
| | | | | $F_{BR} = 360$, | $T_{min} = 249$, | $\gamma = 5$ |
| | | | | *graupel_breakup_* | | |
| | | | | $\aleph_{DS} = 10$, | $p_{max} = 10\%$, | $\sigma = 7$ |

The COSMO interpolation utility (INT2LM) was used to construct initial and boundary conditions from the 7-km COSMO-EU operational assimilation cycle analyses. All simulations are done at 2.8 km spatial resolution with a time step of 25 seconds, 50 vertical levels, and half-hourly output resolution. The Aerosol and Reactive Trace gases module (ART) is turned off. The Phillips et al. (2008) parameterization (PDA08) is used for primary ice nucleation, and the intermediate CCN level of the Segal and Khain parameterization is applied (Segal and Khain, 2006). Previous studies have noted that limited nucleating efficiencies in the PDA08 may lead to underestimation of ICNC at mixed-phase conditions (Barahona et al., 2010; Curry and Khvorostyanov, 2012; Morales-Betancourt et al., 2012). No ice nucleating particle (INP) measurements were made during this case study, but from other observational datasets, PDA08 still yields better agreement with in-situ ICNCs than purely lab-based or theoretical parameterizations (Sullivan et al., 2016). Crosier et al. (2014) also note that the low cloud top temperatures and strong updrafts in convective regions generate supersaturations that could favor large ice production from homogeneous nucleation. While not observationally confirmed, these conditions could buffer the ice nucleation tendency to our choice of parameterization.

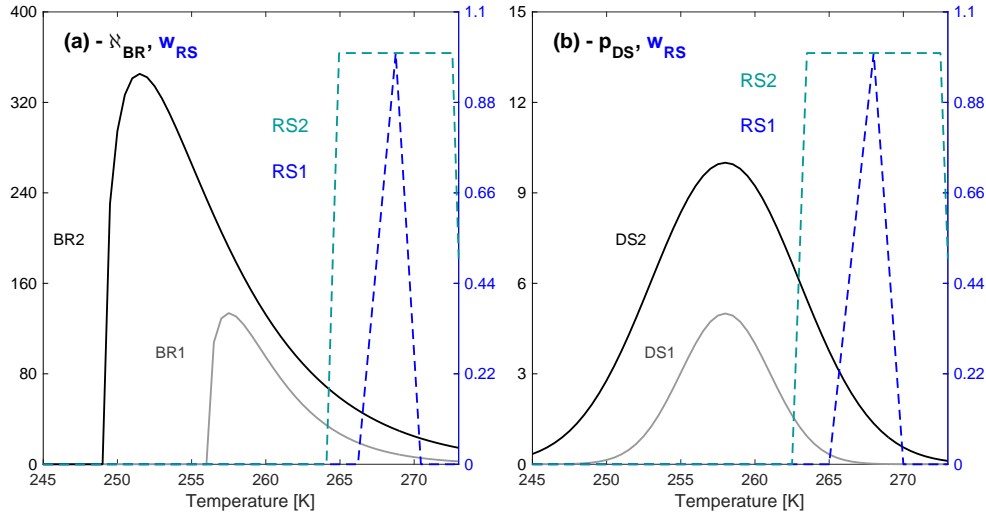

**Figure 2.** Fragment numbers, weightings, and probabilities from the secondary ice production parameterizations. In panel a, we show $\aleph_{BR}$ from both ice-ice collisional breakup simulations (BR1 and BR2) as well as the triangular and uniform $w_{RS}(T)$. In panel b, we show $p_{DS}$ from both droplet shattering simulations (DS1 and DS2) and $w_{RS}$ once again. The overlapping temperature regions of these are particularly important to understand any feedback between the processes.

The rainband structure allows us to investigate multiple secondary production processes at the same time because it contains regions with varying dynamic and thermodynamic conditions. For example, large liquid water contents and stronger updrafts are favorable for frozen droplet shattering (Lawson et al., 2015; Taylor et al., 2016), and these conditions can be found in the narrow leading edge. For rime splintering, lower updrafts and liquid water contents and bimodal droplet size distributions are favorable (e.g. Choularton et al., 1980; Mossop, 1985; Heymsfield and Willis, 2014), and these conditions are found at the top of convective turrets and in the trailing region of stratiform precipitation. The rainband vertically spans the optimal temperature zones for both droplet shattering and rime splintering (Hallett and Mossop, 1974; Leisner et al., 2014; Lawson et al., 2015) and horizontally spans both maritime and continental zones to compare any impact of different surface heat budget.

## 4 Results

### 4.1 Dynamic environment

We begin by comparing the observed and modeled dynamics to understand how these differences may impact later microphysical ones. We show the updraft along a 255° radial out from the CFARR station (Figure 3) as in Figure 5d of Crosier et al. (2014). Comparison to simulations cannot be exact, as the output from the model exists along a coarser spatiotemporal grid. We use those $w$ values whose latitude-longitude pair minimizes the Euclidean distance to the precise values along the radial. Thereafter we interpolate both over distance and altitude on a 1000 x 500 grid to generate the range-height indicator (RHI)-

type plot. This interpolation leads both to intermittent discontinuities and to weakening of extremes. The upright updraft region about 60 km from CFARR appears distinctly in the simulated field but with vertical velocity magnitude far smaller (and extent far greater) than those derived from measurements (maximum of about 1 m s$^{-1}$ relative to about 6 m s$^{-1}$). Downdrafts of more similar magnitude and extent to those observed also form in adjacent regions. Values derived from Doppler velocities ($v_u$) also rely on an assumption that at low elevation, $v_u$ approximates the horizontal wind and that any convergence or divergence of these horizontal winds within discrete columns must conserve mass with a compensating upward or downward velocity.

Comparison of the surface wind speed both in the CTRL simulation and from a three cup anemometer at CFARR is also shown in Figure 3b. Three series are shown from the simulation at latitude and longitudes closest to the center. Simulated wind speed peaks too early but to a value only slightly less than the average of the observations. Both series display a sudden drop in the strength of these winds with similar decay rates and plateau values of about 5 m s$^{-1}$. Perhaps most important is the consistent underestimation of these surface winds prior to the rainband event, from 17:00 to 19:00 UTC. Given that the direction of low-level winds preceding the rainfall event was southwesterly (Crosier et al. (2014), their Figure 4b), underestimating their magnitude will diminish the oceanic moisture advection and moisture source ultimately available to form rain over the continent.

Figures 3c and d also show the vertical velocities in the CTRL simulation at altitudes of 1 and 7 km at 18:00 UTC as the rainband reaches land. Its structure is apparent in the low-level updrafts of about 1 to 2 m s$^{-1}$ (although these are again much weaker than those from observations) and their adjacent downdrafts with similar magnitudes of opposite signs. Elsewhere values are $\pm$ 0.2 m s$^{-1}$ with slow descent presiding. For the upper-level field that corresponds to cloud top, the highest ascending motions also occur around the rainband region and slow ascent ($\leq$ 0.4 m s$^{-1}$) dominates.

We next compare range-height indicator (RHI) scans of radar reflectivity ($Z_{DR}$) from the Chilbolton Advanced Meteorological Radar (CAMRa) and the CTRL simulation (Fig.4). The CAMRa is a 3 GHz Doppler instrument with a 0.28° beam, and its scan between 19:22:07 and 19:23:07 UTC along the 255° radial out from CFARR is shown, as in Figure 5a of Crosier et al. (2014). We use output from 19:00:00 UTC in the CTRL simulation and again identify the modeled latitude-longitude pair that minimizes the Euclidean distance to the exact value from along the 255° radial. We then perform bilinear interpolation on the simulated values of $Z_{DR}$ over a 1000 distance x 500 altitude grid.

The CAMRa scan shows the location of cloud top height and convective activity: the lowest $Z_{DH}$ is around 6 to 8 km and fall streaks are present moving toward the CAMRa. These $Z_{DH}$ fall streaks, as well as those in differential reflectivity (shown in Crosier et al. (2014), their Fig. 5c) are signatures of ice crystal sedimentation and aggregation near cloud top. Ice crystal seeding may also be occurring with lower-level sedimentation, but the altitudinal peak in $N_{i,pri}$ does not fall consistently above that in $N_{ice}$ (Figs. S3 and S4) and secondary ice production must also generate a portion of this low-level ice. Intermediate values of $Z_{DH}$ occur at altitudes of 2 to 5 km, and the highest ones occur around the melting layer at 1 to 2 km, as discussed by Crosier et al. (2014). General features are replicated in the simulated reflectivities. Very low $Z_{DH}$ occur close to CFARR with cloud top around 7 km, but further out – around 70 to 100 km along the radial – these same very low reflectivities occur more often than in the measurements. The gradient to higher $Z_{DH}$ at lower altitudes is also apparent in simulations, but not as distinct fall streak structures. $Z_{DH}$ has increased to about 10 dBZ$_h$ by a height of 4 km and about 20 dBZ$_h$ by a height of 2

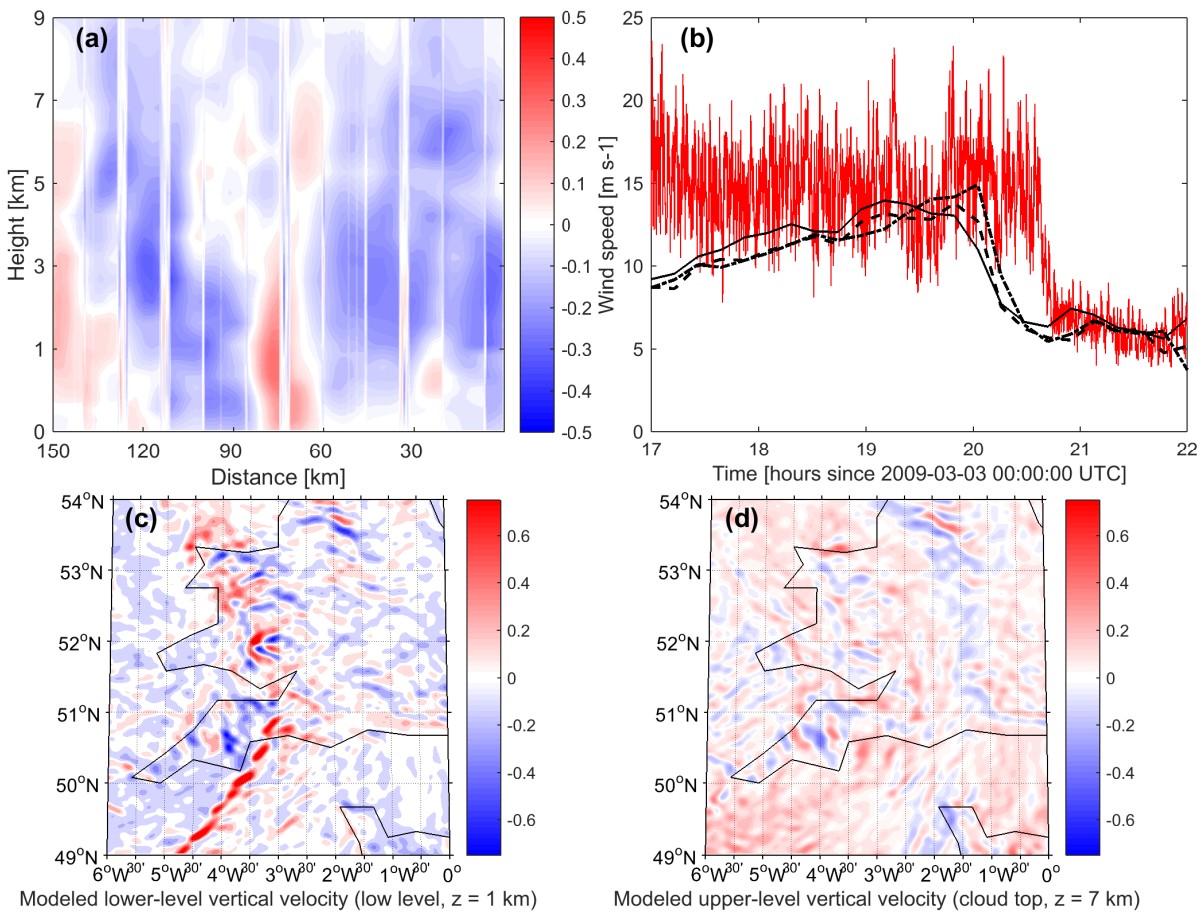

**Figure 3.** Different model dynamical fields. In panel a, we show the updraft velocity 150 km along the 255° radial from CFARR at 51.1°N, 1.4°W with values from the CTRL simulation at 19:00 UTC. Discontinuities are due to the minimization of Euclidean distance or interpolation aspects of an algorithm to approximate the radial from the model discretization. Surface wind speeds from 17:00 to 22:00 UTC from the CFARR three cup anemometer and our CTRL simulation are shown in panel b. Panels c and d show the modeled vertical velocities from the CTRL simulation at 1 and 7 km respectively at 18:00 UTC as the rainband began to pass over the UK.

30   km. The highest reflectivities also fall in the same altitudinal range, but importantly do not have the same maximum as in the observations. $Z_{DH}$ in an updraft core 60 km from CFARR reaches a value of 45 dBZ$_h$ in the CAMRa but only 30 dBZ$_h$ in the CTRL simulation. This may be due to underestimation of graupel formation or too high CCN or INP concentrations that delay precipitation in the base COSMO model (e.g. Baldauf et al., 2011). In Figure S1, we also show radar comparisons for the simulations with secondary ice parameterizations in place (RS1, DS1, BR1ig). These exhibit $Z_{DH}$ of lower magnitude than the observations, even more so in the first 50 km extending from the CFARR site. We keep these underestimation in mind in the proceeding discussion of microphysical adjustments.

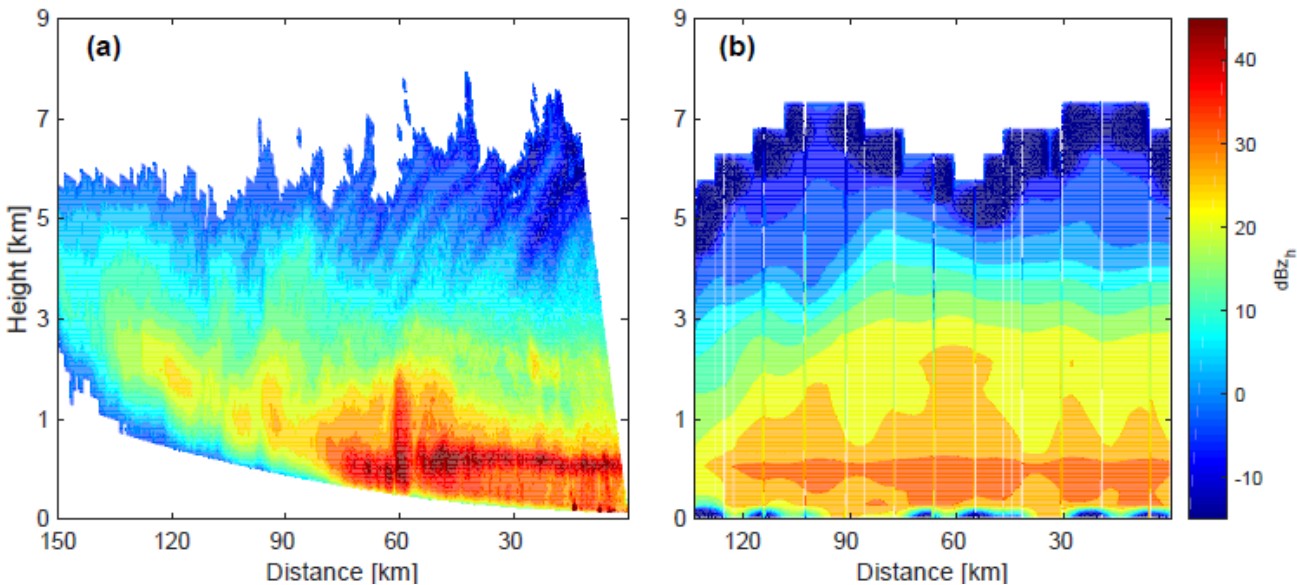

**Figure 4.** Model-measurement intercomparison of range-height indicator scans of radar reflectivity $Z_{DH}$ along the 255° radial out from CFARR. CAMRa Doppler radar measurements are shown in panel a for the scan taken between 19:22:07 and 19:23:07 UTC, and modeled values are shown from the CTRL simulation at 19:00:00 UTC, both in dBZ$_h$.

### 4.2   Ice crystal production

We first present the spatial fields of secondarily-produced ICNC, $N_{i,sec}$, and the corresponding fields of primarily-nucleated
ICNC, $N_{i,pri}$, at a single level in Figure 5. An altitude of 3 km is chosen so that the temperature falls in the rime splintering temperature range ($T \in$ [249 K, 270 K] with a median value of 258 K), and we show first the rime splintering simulations. $N_{i,sec}$ is an accumulated ICNC from the secondary ice production between 18:00 and 18:30 UTC, as the cold frontal rainband begins to pass over the UK, and does not include any sedimentation loss to lower levels or gain from higher ones. In addition, we have filtered out all values less than $10^{-3}$ L$^{-1}$ to give a clearer signal of where significant production occurs.
ICNC of up to 1000 L$^{-1}$ are produced over the half hour and over much of the domain. Banded structures appear in $N_{i,sec}$ off the southern coast of the UK, where the largest raindrop number concentrations (Fig. S5d) and the highest updrafts at lower

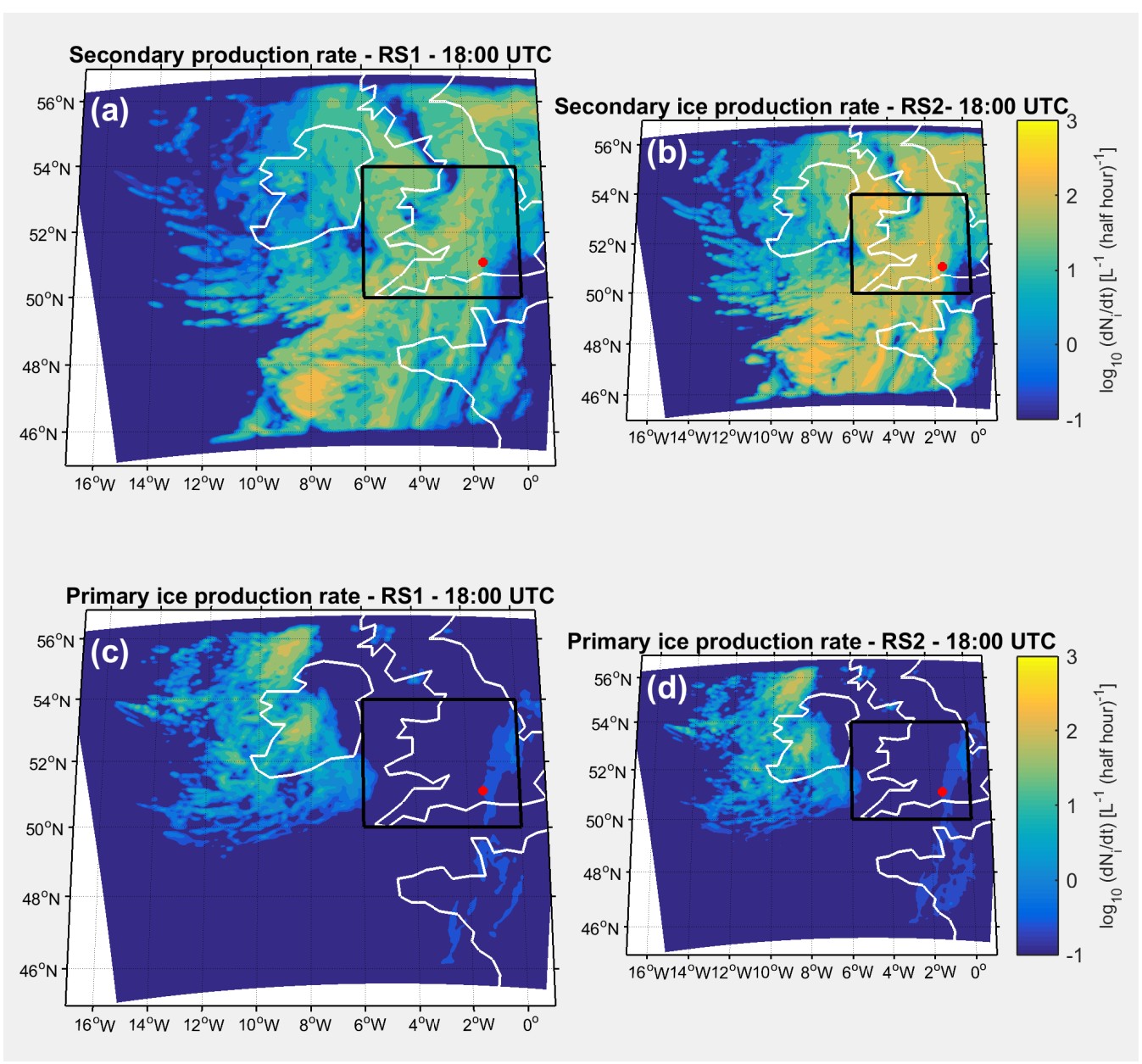

**Figure 5.** Maps of secondarily-produced ice from the RS1 (on the left) and RS2 (on the right) simulations at the altitude where rime splintering contributions are larger. Temperatures at this altitude fall between 249 K and 270 K with a median value of 258 K. Panels a and b show the number of secondarily-produced ice crystals from 18:00 UTC to 18:30 UTC, and panels c and d show the primarily-nucleated ICNCs for the same time period, as the rainband began to pass over the UK. Note the logarithmic colorbar. The black box shows the subdomain used in Figure 7, and the red dot is the CFARR location.

levels (Fig. 3c) occur. Underestimation of the updrafts within the rainband core (Sec. 4.1) will lead to errors that offset each other somewhat: too few raindrops will form when the vertical velocity and supersaturation are too low, but these will also be lofted more quickly through the altitudinal band where rime splintering is favorable, leaving less time for collisions.

In spatial maps of graupel and snow mixing ratios (Fig. S5), and to a lesser degree in panels a and b, the characteristic comma shape of clouds in a well-developed mid-latitude cyclone is apparent. Although graupel was absent throughout much of the observations, simulated graupel mixing ratio is non-negligible (greater than $0.25$ g kg$^{-1}$) at low altitudes coincident with the rainband. The snow mixing ratio is also from $0.25$ to $0.35$ g kg$^{-1}$ but at higher altitudes. It is likely then that much of the riming occurred on snow, as suggested by Crosier et al. (2014), and fell to lower altitudes with some portion splintering during descent.

    In the RS2 simulation, large values of $N_{i,sec}$ are found further north than in the RS1 simulation. For example, over the
cold waters of the North Sea and Irish Sea, the rime splintering contribution moves to lower altitudes for the more limited temperature weighting $w_{RS}$ in the RS1 simulation. The same is true for much of the continental ice production. In some instances of moisture limitation, we might expect that an extended $w_{RS}$ actually decreases the rime splintering contributions at lower altitudes, as the rime mixing ratio may be depleted above. In this case, however, for a primarily oceanic domain, limited adjacent land, and strong surface winds (Fig. 3b), moisture advection is not limited.

For comparison, we show the corresponding primary ice nucleation fields in panels c and d. Again we filter out where these values are less than $10^{-3}$ L$^{-1}$. Much of the low-level ice comes from secondary production, although the temperatures at this altitude are still low enough for various aerosol to act as INP. In the rainband region, $N_{i,pri}$ is negligible and only in a small region in the northwest of the domain does it reach $100$ to $1000$ L$^{-1}$. The magnitude of these values is subject to uncertainty from the nucleation parameterization, which, as noted above, has underestimated INP numbers in previous studies.

Next, we consider the relative ice crystal number concentrations produced by different processes in Figure 6. The largest $N_{i,sec}$ magnitudes, up to $1000$ L$^{-1}$ over the half hour, come from the RS1 and ALL simulations. These are followed by about $10$ L$^{-1}$ (half hour)$^{-1}$ generation rates from frozen droplet shattering and $0.1$ L$^{-1}$ (half hour)$^{-1}$ of snow and graupel. There is also an altitudinal hierarchy. Contributions from droplet shattering are largest at the highest altitudes of $4.5$ km where raindrop number concentrations are still relatively high and the temperature ($T \in [237$ K, $262$ K$]$ with a median of $249$ K) is cold enough
for non-negligible shattering probability. The rime splintering contribution is next at an altitude of $3$ km, and the breakup is largest at a lower altitude of $1.5$ km because the graupel mixing ratio is highest here.

    If graupel were present at higher altitudes, $N_{i,sec}$ from breakup could increase significantly, as both the snow mixing ratio and fragment number parameter increase at colder temperatures. Similarly very limited raindrop number concentrations exist at the altitude where shattering probability is non-negligible. This importance of large hydrometeor number concentration
for $N_{i,sec}$ suggests a means of parameterizing secondary ice production as meso- and large-scale models transition toward two-moment cloud schemes: the droplet shattering and collisional breakup parameterizations could be activated only for those cloudy grid cells with more than a threshold concentration of large droplets and graupel, hail, or snow respectively. In one-moment schemes, parameterizations on the basis of favorable thermodynamic regimes will be more useful for the time being.

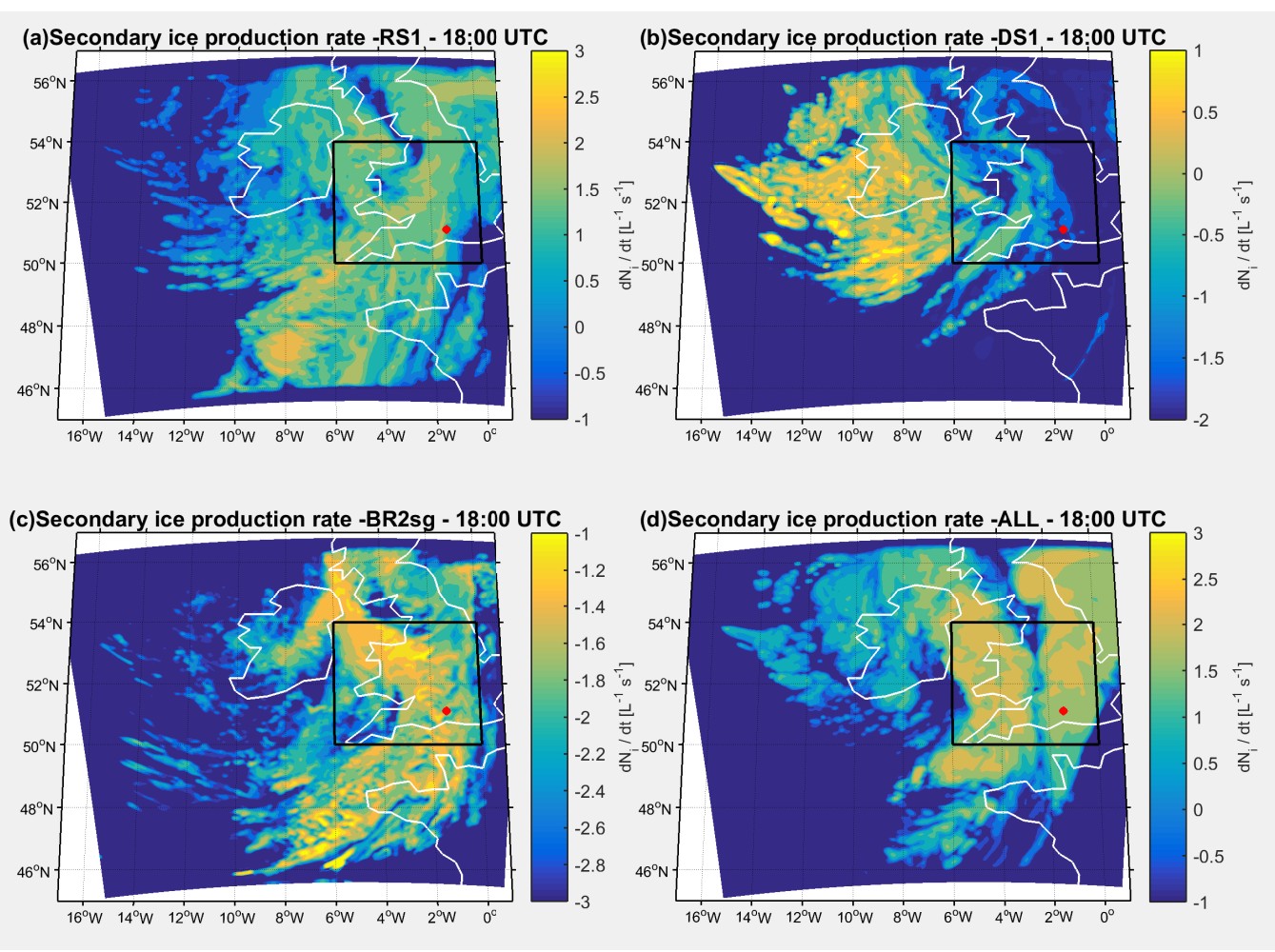

**Figure 6.** Maps to compare the secondarily-produced ice crystal numbers between 18:00 UTC and 18:30 UTC from (a) rime splintering (RS1 at 3 km) (b) frozen drop shattering (DS1 at 4.5 km) (c) collisional breakup between snow and graupel (BR2sg at 1.5 km) and (d) all secondary processes occurring simultaneously (ALL at 4.5 km). Note the different logarithmic colorbars for each panel. The black box shows the subdomain used in Figure 7, and the red dot is the CFARR location.

Along with these spatial fields, we show the altitudinal, probability, and temporal distributions of modeled and observed ICNC in Figure 7. In panel a, we compare the altitudinal profiles of ICNC from 18:00 to 21:00 UTC and in a sub-domain centered at CFARR ($49°$ to $54°$N latitude and from $6°$W longitude to the Prime Meridian as shown in Figs. 5 and 6). Modeled values were output for altitudes from 1.5 to 7 km hPa with 250 m intervals for the first two and 500 m thereafter. Observations from a cloud imaging probe CIP-15 are discretized into the largest number of altitude bins that still give reliable statistics. IAT algorithms, with those particles below a threshold IAT of $10^{-4}$ s classified as artifacts, were used to correct the ICNCs Field et al. (2006). No shatter-resistant tips were used on the probe, but given the strict IAT threshold, the possibility of artifacts is limited. The envelopes on all profiles represent the first and third quantiles of ICNC in the interval or bin. Simulated values less than 1 m$^{-3}$ are filtered out.

At the higher altitudes and colder temperatures, the modeled ICNCs all overlap because they use the same representations of heterogeneous and homogeneous ice nucleation. Both the simulated median and standard deviation are also in good agreement with the CIP-15 observations, except for slight underestimations around 6 to 7 km. At the lowest altitudes and warmest temperatures, however, the ALL simulation is the only one whose envelope overlaps the observations, both because of the larger rime splintering temperature weighting and co-occurrence of all secondary production processes. Taylor et al. (2016) also found in their study of cumuli in the same Southwestern Peninsula region that combinations of secondary production processes were necessary to explain observed ICNCs. The importance of the graupel at low levels can also be seen by the high $N_i$ at the lowest level of 1.5 km (as in Fig. 6c). The underestimated updrafts and radar reflectivities noted above in Section 4.1 may also help explain the too low $N_i$ around 2 km: larger vertical velocities could loft graupel to high altitudes and boost the contribution from collisional breakup. A final contribution to these too low ICNCs may be inclusion of parts of the warm front in the subdomain of analysis.

In panel b, we show the ICNC probability distributions from cloud probe observations (in black) and the CTRL, RS1, DS1, BR1ig, and ALL simulations (in color) between 18:00 and 21:00 UTC and between 1.5 and 3.5 km. Values come again from the subdomain centered at CFARR (shown in black in Figs. 5 and 6). Although the values of less than $10^{-3}$ L$^{-1}$ have been filtered out, the proability of small $N_{ice}$, less than 1 L$^{-1}$, is still much higher for all simulations than observations. At these lower concentrations, however, measurements will be less precise because of sampling volume and flow rate limitations, and model output is more susceptible to numerical noise. The observed distribution is more skewed than the simulated ones with the exception of the ALL simulation: it extends out to $N_i$ of 100 L$^{-1}$ with probabilities of 0.1%. Interestingly, the RS1 simulation that produces the largest $N_{i,sec}$ in the single process simulations (Fig. 6) underestimates the intermediate values of $N_i$ between about 5 and 20 L$^{-1}$.

The simulated and observed distributions vary strongly in their higher order moments. The observed distribution is far more skewed with a long tail out to $N_{ice}$ of 100 L$^{-1}$ and has a higher kurtosis as the larger probabilities at low $N_{ice}$ drop off quickly. The simulated distributions drop off more quickly with negligible probabilities by about 45 L$^{-1}$, and their kurtosis is larger, as the high probabilities at low $N_{ice}$ drop off only slowly through the intermediate values. The simulation distributions themselves overlap strongly, but differences are present at 35 L$^{-1}$ and above. By concentrations of 45 L$^{-1}$, only the RS1 and

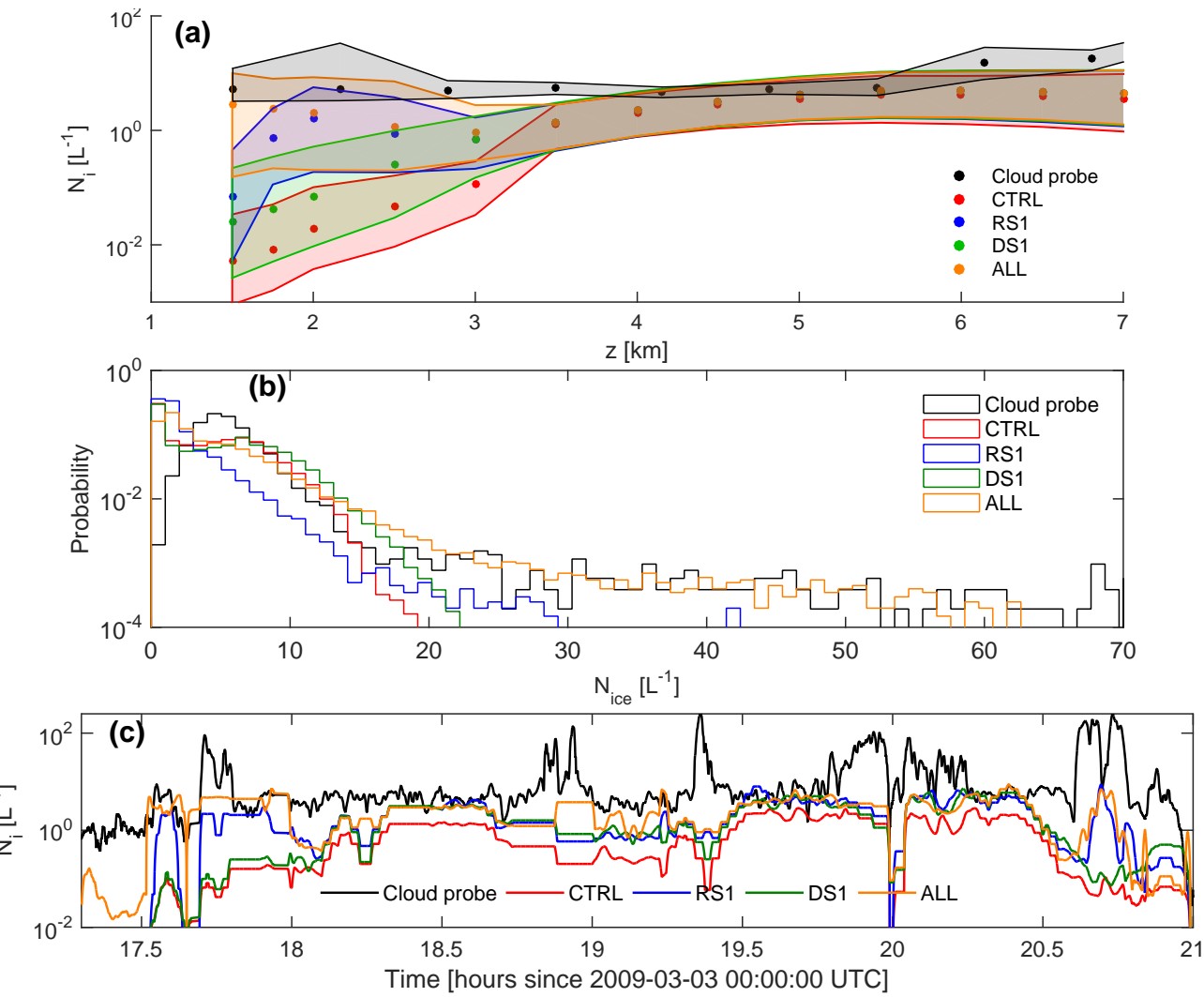

**Figure 7.** Various distributions of APPRAISE CIP-15 observed and simulated ICNC between 17:00 and 21:00 UTC for a small domain centered at CFARR with logarithmic scales. Panel a shows altitudinal profiles, where the dotted values indicate median, while the upper and lower envelopes show the first and third quantiles. Panel b shows probability distributions and panel c shows the 30-point running mean time series. Euclidean distance is minimized to find simulated values that correspond to the observed ones in the time series.

ALL simulations have non-negligible probabilities. The CTRL simulation is the only one for which all probabilities of $N_i$ greater than $20\,\mathrm{L}^{-1}$ are negligible.

Finally, we look at the temporal evolution of $N_{ice}$ in both observations (in black) and the CTRL, RS1, DS1, and ALL simulations (in color) in panel c. To collocate a simulated value with the observed one at a given time, we use the grid cell that minimizes the Euclidean distance to the latitude, longitude, and altitude where the measurement was made. In some instances then, for example between about 18:15 and 18:35 UTC, a simulated value is reused because the aircraft circled the same location. Thereafter a thirty-point running mean is calculated for all series. The underestimate in the simulations is apparent again, although it is the least for the ALL simulation, as in the altitudinal profiles of panel a. The observed trends – sudden increases around 17:30 UTC and after 20:30 UTC or the sudden decrease at 20:00 UTC– appear also in the simulated time series. But the ICNC magnitude is always closest to observations in the ALL simulation followed by RS1 and then DS1. The

CTRL simulation always has the largest discrepancy from cloud probe values. So although simulated ICNC remains too low, the secondary ice parameterizations do consistently shift the values in the right direction.

## 4.3   Impact on precipitation

Changes in the spatial and temporal distributions of ICNC are expected upon addition of another ice generation mechanism. But the new parameterizations may also have an indirect impact on other metrics. In particular, we consider adjustments to the

spatial distributions of accumulated precipitation $P_{tot}$ and of precipitation rate $\dot{P}$. In Figure 8, $P_{tot}$ is shown for a small domain near CFARR at 23:30 UTC, after full passage of the rainband over the UK. Panel a shows $P_{tot}$ from the CTRL simulation, which reaches $30\,\mathrm{mm}$. Larger accumulated rainfall over continental than maritime environments is consistent with observations of other narrow cold frontal rainbands (NCFR) (Viale et al., 2013). In panels b through f, deviations from the CTRL simulation are shown and are largest in those regions where the magnitude of $P_{tot}$ is already large. With magnitudes of up to $10\,\mathrm{mm}$,

these deviations represent 30% of the signal. The sum of the deviations over the whole subdomain is an additional $23.9\,\mathrm{m}$ of precipitation for the RS1 simulation, $25.6\,\mathrm{m}$ for RS2, $16.9\,\mathrm{m}$ for DS1, and $16.6\,\mathrm{m}$ for ALL. While the combination of processes generates more ice, it also produces less total precipitation than the secondary prodution processes alone. We emphasize that these $P_{tot}$ deviations are calculated using an average of 5 ensemble runs, so that the signal reflects the change in microphysics not numerics.

Regions of positive $P_{tot}$ deviation are spatially followed by negative $P_{tot}$ deviation and vice versa. This banding appears in the absolute $P_{tot}$ field of panel a as well and reflects convective structure: vertical motion is strongest in the rainband leading edge, but also preceded and proceeded by downdrafts (Fig. 3c). Interestingly, additional ice generation increases the magnitude of these oscillations in convective precipitation. This amplification may be due in part to more riming in the ascending regions, which feeds into precipitation both directly as rimed particles sediment and melt (Fig. 1*(3)*) and indirectly as they splinter and

generate more rimable particles (Fig. 1*(4)*). Higher concentrations of rimed particles would be in line with the higher values of differential reflectivity at the convective cloud top, noted by Crosier et al. (2014). Orography in this region also has an impact. The spot of particularly large $P_{tot}$ around 50.5°N and 4°W corresponds to the Dartmoor with a maximum elevation of $621\,\mathrm{m}$. Slightly elevated $P_{tot}$ is also present over the Exmoor and Bodmin Moor at (51°N, 3.5°W) and (50.5°N, 4.5°W) respectively.

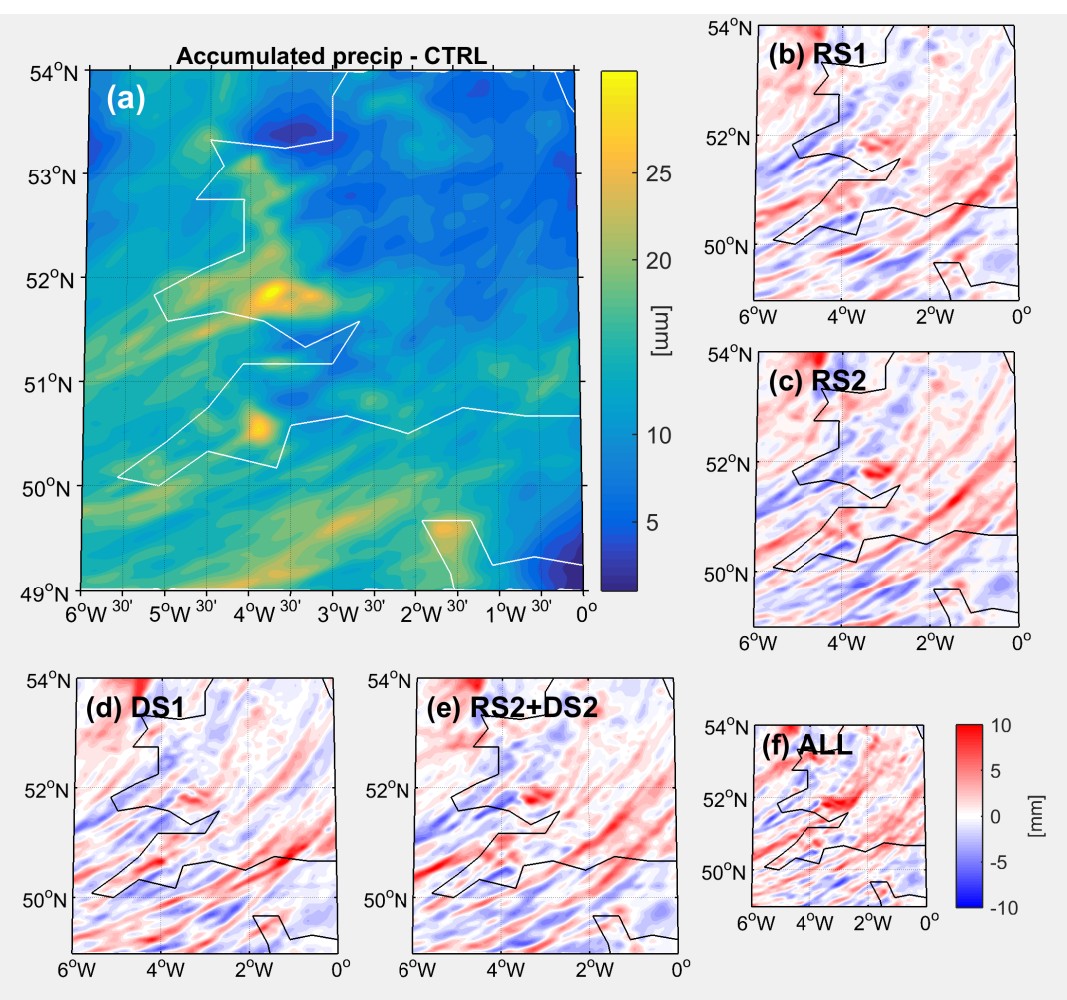

**Figure 8.** Panel a shows the spatial distribution of accumulated precipitation in the control simulation for a small domain near CFARR at 23:30 UTC, after full passage of the rainband over the UK. Panels b though f show the deviations of accumulated precipitation in five of the simulations with the secondary ice parameterizations in place.

As in Figure 5, the impact of adjusting $w_{RS}$ can be seen here between RS1 in panel b and RS2 in panel c. The increased temperature weighting actually reduces the magnitude of $\Delta P_{tot}$ in some regions for the RS2 simulation. As in the ALL simulation above, this could be due to the kind of 'cloud lifetime effect' shown in Figure 1(1) in which many small ice crystals do not sediment as quickly. Deviations in the DS1 precipitation are concentrated in a single band and are reduced over the continent where the large droplets are less numerous relative to those from the RS* simulations. The largest changes in $P_{tot}$ are actually due to modest rime splintering.

Deviations in $\dot{P}$ (Fig. S2) exhibit much of the same behavior as the $P_{tot}$ field. The highest and lowest deviations align almost exactly with the most extreme low-level updrafts (3). 'Hotspots' appear again in the $P_{tot}$ field with 'coldspots' in between. Here the impact of the updraft structures is even more evident, with the 'hotspots coincident with cores of convective precipitation in the NCFR and surrounding 'coldspots' coincident with its more stable gap regions (James and Browning; Hobbs and Persson, 1982; Crosier et al., 2014). The addition of the secondary ice parameterizations both invigorates the precipitation in the convective regions and strengthens the corresponding drying in adjacent gap regions.

Finally in Figure 9, we show the simulated and observed time series of precipitation rate in color and black respectively for the same domain near CFARR between 17:00 and 22:00 UTC. Observations come from the Met Office C-band NIMROD System of rain radars at 5 km spatial and 15 min temporal resolution. The domain-mean values and standard deviation envelopes are shown for both. In panel a, the standard deviation of mean values in a $1°$ longitude-$1°$ latitude box (1-2°W, 51-52°N) around CFARR is used, whereas in panel b, the mean standard deviation over the 5-simulation ensemble is used. The first can be understood as spatial variance and the second as numerical, and the first is much larger. Because the rainband is so narrow, this is especially true at later times during its passage: the intense precipitation may extend over only 10 kilometers or less, while the small domain around CFARR is still about 50 kilometers by 50 kilometers. In neither case does the evolution of the mean $\dot{P}$ for different simulations vary significantly from one to the next.

Spatial variance aside, the simulations do not capture the rainband narrowness well in the time series. There is a broad increase in precipitation rate from about 19:00 UTC through 21:00 UTC, but not the sharp increase to 40 mm h$^{-1}$ as in the observations. The simulations reproduce at most 75% of the magnitude of the maximum $\dot{P}$ but tend to fall within the spread of the observations. This similarity of the simulated series upon spatial averaging may be expected because of the finer-scale rainband structure discussed above, but the averaging is retained for a more robust trend. And in spite of this, the mean $\dot{P}$ remains closest to the observations in the RS2 simulation, followed by the RS1 and DS1 ones.

## 5   Discussion

Additional ice generation in different regions, as shown in Figures 5 and 6, will have energetic implications. First many small ice crystals are more likely to be advected into a convective outflow region where they will contribute to cirrus optical thickening and resultant surface warming. As mechanical processes, rime splintering and collisional breakup do not have direct latent heating effects; however, droplet riming becomes more likely in the presence of higher ICNCs, creating an 'indirect latent heating effect'. For droplet shattering, the phase change as the raindrop freezes releases heat of fusion and may invigorate the

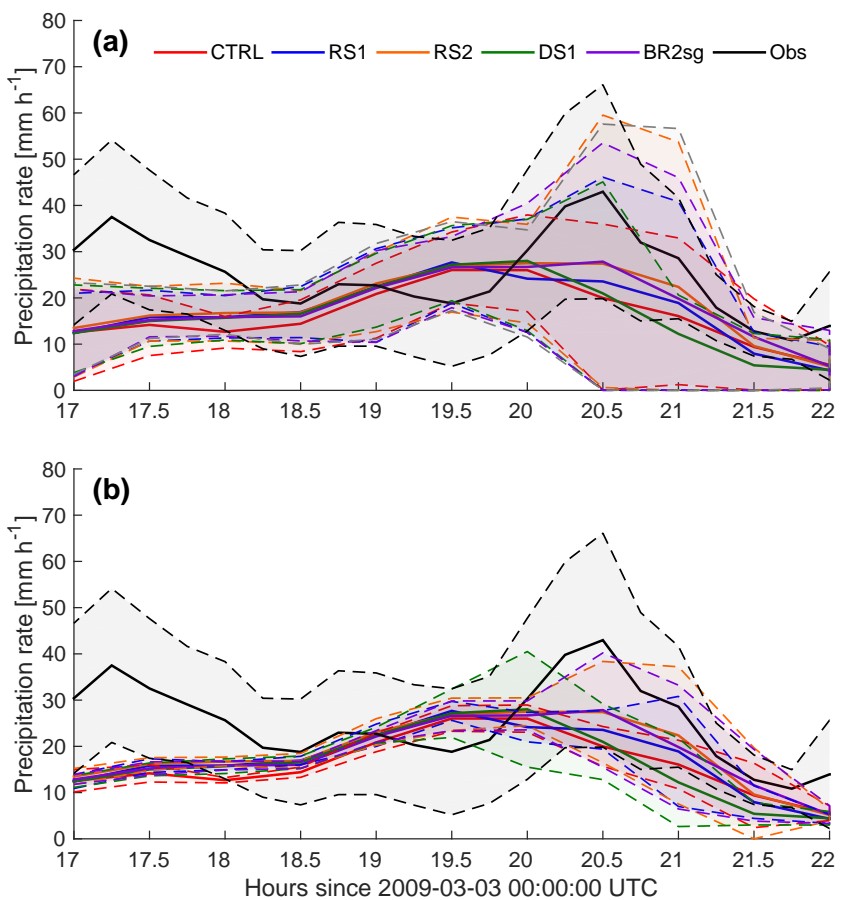

**Figure 9.** Time series of precipitation rate, both from the UK NIMROD rain radar in black and from various simulations in color for a small domain around CFARR from 17:00 to 22:00 UTC. For all traces, the mean is shown with an envelope of plus-minus a standard deviation for the observations and the first and third quantile for the simulations. In panel a, the quantiles are calculated over the small spatial domain, and in panel b, they are calculated over the 5-member ensemble.

surrounding updrafts. More latent heating aloft can intensify the upper-level anticyclonic potential vorticity (PV) (Willison et al., 2013). This PV generation may suppress further cyclogenesis by disconnecting the developing system from surface potential temperature anomalies. On the other hand, higher cyclonic diabatic PV may slow system progression and maintain favorable levels of shear. Freezing should also intensify detrainment, as shown in the modeling sensitivity studies of Sud and

25  Walker (2003). In these mechanisms, ice crystal production can be a mediator of circulation-radiation feedbacks.

Variations of aerosol concentration and surface radiative budget over continent versus ocean should also affect the efficiency of these processes. As aerosol loadings increase over continental regions, more and smaller cloud droplets form for a fixed cloud liquid water content. This shift in the droplet size distribution will diminish the riming and droplet freezing efficiencies. In this case, however, without moisture limitations and only a narrow continental region, no notable land-sea differences appear

30  in our spatial snapshots (e.g., Fig. 5).

From Figures 5 and 6, the model can generate 1000 crystals per liter per half hour by secondary production. In mixed-phase parcel model simulations, we have calculated slower but longer-lasting secondary ice production rates than in these mesoscale simulations (Sullivan et al., 2017, 2018). In simulations that include rime splintering and ice-ice collisional breakup, the model generates about 1 to 10 $L^{-1}$ (half hour)$^{-1}$ even when a description of ice non-sphericity is included. Larger production rates in the mesoscale model can be attributed in part to the representation of the largest hydrometeors, which are crucial both to secondary ice production and precipitation processes. The parcel model contains only six size bins and so does not capture the tail of the size distribution present in the COSMO simulations. Inclusion of hydrometeor number from these tails will enhance the tendencies in Equations 1, 4, and 7. But it also enhances the sedimentation rates, so that that multiplication feedbacks (like

that shown in green in Figure 1) will be limited in their efficiency.

We have also used parcel model simulations to comment on the relative importance of droplet shattering and ice-ice collisional breakup. Sullivan et al. (2017) found that the dominance of rime splintering versus collisional breakup was determined by timing of large hydrometeor formation in the liquid versus ice phase. Given the relatively low graupel and hail numbers for this cold frontal rainband (Crosier et al., 2014), it is to be expected then that the rime splintering parameterization is most

influential on $P_{tot}$ (Section 4.3). The analogous $N_{i,sec}$ field for ice-ice collisional breakup (Figure 6c) is about four orders of magnitude less than those for rime splintering (Figure 5). These lower magnitudes reflect to some extent the "self-limiting nature" of ice-ice collisional breakup as well because ice hydrometeors must be consumed to generate them in this process. Localized enhancement and dynamical dependence of these secondary production processes should be even more evident for simulations at higher spatial resolutions: convection would be better resolved and higher liquid water contents would be

generated.

The spatiotemporal resolution, along with the microphysics scheme, also plays into the strong feedbacks between hydrometeor formation, latent heating, and cloud dynamics. Previous studies have generally found a spatial resolution of 4 to 6 $km$ to be sufficient to reproduce precipitation extremes (e.g. Prein et al., 2013; Pieri et al., 2015). This resolution dependence results from changes in the vertical moisture advection, in turn due to adjustments of vertical velocity with resolution (Yang et al.,

2014). For simulations whose resolutions border on the "gray zone" scales (around a tenth of a degree), over-representation of convective activity is possible by both the parameterization and explicit resolution (Pieri et al., 2015); however, our simulations

are at a fully convection-permitting scale and use only a reduced form of the Tiedtke mass-flux scheme for shallow convection (Tiedtke, 1989) and this should not be a concern.

The use of a two-moment scheme is also important for simulation of extreme precipitation (Otkin et al., 2006). Certain one-moment schemes tend to generate overly large drops and too high precipitation rates (Thompson et al., 2004), but SB06 tends to produce especially large quantities of graupel (Otkin et al., 2006). The Bigg parameterization, as a precursor to our droplet shattering additions, has been shown in previous studies to predict very low numbers of frozen drops (e.g. Morrison et al., 2005; Fan. et al., 2009), which may contribute to underestimation of its contribution here. The more rigorous alternative would be to account for immersed surface area and scavenging of ice nucleating particles as in Paukert and Hoose (2014); Paukert et al. (2017), and future work should implement an updated immersion freezing parameterization.

## 6 Summary

New frozen droplet shattering and ice-ice collisional breakup parameterizations have been developed and implemented into the regional COSMO weather model. We performed several simulations of a cold frontal rainband, observed during the APPRAISE campaign over the UK, with adjustments to the new parameterization formulations. With these runs, we have shown the following:

1. *ICNC generated by secondary production processes can be as large as those from primary nucleation, especially in the presence of sufficient convective activity and rain, snow, or graupel mixing ratios.*

   The new parameterizations calculate $N_{i,sec}$ production rates of up to $1000 \, \text{L}^{-1}$ per half hour, values on the order of the primary nucleation tendency at certain altitudes. The localized regions of large secondary ice production are characterized by convective updrafts and by optimal temperatures that maximize parameters like $w_{RS}$ or $p_{DS}$. The role of the convective updraft is two-fold, both to generate a supersaturation that forms and grows large hydrometeors and then to loft those to the optimal temperature zone. In this case, we saw that rime splintering was the most important process in line with the conclusions of Crosier et al. (2014); however, underestimation of vertical velocities in the cold front also led to underestimation in simulated radar reflectivity relative to observations. If this $Z_{DH}$ difference was caused by additional graupel at higher altitudes, contributions from collisional breakup could have been much higher than the $0.1 \, \text{L}^{-1}$ per half hour found here. A low bias in updrafts also generates fewer raindrops at altitude and limits the contribution from frozen droplet shattering (in this case to an intermediate production rate of $10 \, \text{L}^{-1}$ per half hour).

2. *ICNCs greater than $50 \, L^{-1}$ are underestimated with the addition of single secondary production parameterizations.*

   Generally, the addition of secondary ice parameterizations moves the ICNC magnitudes in both vertical and temporal distributions toward the observed values, but underestimates persist, particularly at altitudes between about 2 and $4 \, \text{km}$. In time series, sudden increases or decreases in ICNC are well-captured but their magnitude remains about an order of magnitude too small. The combination of processes in the ALL simulation does the best in reproducing the observed histogram and time series, but a feedback between the processes may still be missing in the current formulations. Under-

estimations stem in part from low biases in the updraft velocity. If the vertical velocities can be brought into agreement with observations, then criteria in these values as well as temperature could be used together to parameterize secondary production in appropriate thermodynamic zones. For two-moment schemes, graupel, snow, and raindrop criteria could be implemented for these processes

3. *The variation between convective precipitation regions and neighboring quiescent ones is intensified by the addition of secondary ice production.*

The addition of secondary ice systematically increases accumulated precipitation or precipitation rate in regions where these values are largest. The magnitude of these increases can be up to 30% of the signal, again in very localized regions, and reduction in $P_{tot}$ and $\dot{P}$ occur in adjacent gap regions. For this particular case, rime splintering is the most influential process for precipitation formation, and interestingly, widening its temperature weighting actually weakens the positive anomalies of $P_{tot}$ in a kind of 'cloud lifetime effect' whereby additional small ice generation lengthens cloud duration. Inclusion of secondary ice parameterizations does not bring the simulated maximum precipitation intensity much closer to the observed value, and the narrowness of the rainband is still not well-reproduced.

# 7 Code availability

Documentation of the updates to SB06, the Seifert and Beheng two-moment scheme within the larger COSMO model, ice-ice collisional breakup subroutines, and NetCDF4 output files from sensitivity tests are all available upon request.

# 8 Data availability

The UK 1 km-resolution NIMROD radar data are available through the Met Office Centre for Environmental Data Analysis at http://catalogue.ceda.ac.uk/uuid/82adec1f896af6169112d09cc1174499. Filtered CIP-15 ICNC data from the 3 March 2009 flight are available upon request, and their corresponding latitude, longitude, and altitudes are available through CEDA at http://data.ceda.ac.uk/badc/faam/data/2009/b433-mar-03/core_processed.

# Appendix A:  Rain freezing parameterization

Within the Seifert and Beheng two-moment scheme, the number of freezing droplets is calculated with the stochastic model of Bigg (1953):

$$\ln(1 - P_{freez}) = - \int_0^t V_R J_{het} dt, \tag{A1}$$

$$= -\frac{\pi}{6} \int_0^t D_R^3 J_{het} dt \tag{A2}$$

where the heterogeneous nucleation coefficient is defined as $J_{het} = b_{het}\exp(a_{het}\Delta T - 1)$ and $\Delta T$ is the supercooling. We use $a_{het}$ of 0.65 and $b_{het}$ of 200 as in Barklie and Gokhale (1959). Then the number of freezing droplets per time is given by

$$\frac{\partial N_{freez}}{\partial t} = -N_R \bar{x}_R J_{het} \tag{A3}$$

where $N_R$ is the raindrop number and $\bar{x}_R$ is the mean mass per raindrop.

## 5 Appendix B: Hydrometeor size distributions and collision integrals

The generalized $\Gamma$ size distribution is used for the hydrometeor size distributions within the Seifert and Beheng two-moment scheme:

$$f(x) = Ax^\nu \exp(-\lambda x^\mu) \tag{B1}$$

where $(\nu + 1)/\mu$ is the shape parameter, $\lambda$ the rate parameter of the distribution, and $x$ is hydrometeor mass. Then the number
of particles of type $j$ that are collected by type $k$ is given by

$$\left.\frac{\partial N_j}{\partial t}\right|_{coll,jk} = -\int_0^\infty \int_0^\infty f_j(x_j) f_k(x_k) K_{jk}(x_j, x_k) dx_j dx_k \tag{B2}$$

where $K_{jk}$ is the collection kernel, the product of the collisional cross section, a collection efficiency, and a differential settling velocity:

$$K_{jk}(x_j, x_k) = \frac{\pi}{4}[D_j(x_j) + D_k(x_k)]^2 E_{jk}(x_j, x_k)|v_j(x_j) - v_k(x_k)| \tag{B3}$$

To obtain the analytical expression for the collision integral in Equation 6, the collection efficiency is assumed to be independent of particle sizes: $E_{jk}(x_j, x_k) \approx \overline{E_{jk}}$. The particle diameter is assumed to relate to particle mass through a power law expression: $D_j(x_j) = a_j x_j^{b_j}$. And the differential settling velocity is approximated by a characteristic difference: $|v_j(x_j) - v_k(x_k)| \approx \overline{\Delta v_{jk}}$. Then Equation B2 becomes

$$\left.\frac{\partial N_j}{\partial t}\right|_{coll,jk} = -\frac{\pi}{4}\overline{E_{jk}}\,\overline{\Delta v_{jk}} \int_0^\infty \int_0^\infty f_j(x_j) f_k(x_k)[D_j(x_j) + D_k(x_k)]^2 dx_j dx_k \tag{B4}$$

Wisner et al. (1972) have solved this equation with the non-dimensional values $\delta$ and $\theta$ given by

$$\delta_j^\kappa = \frac{\Gamma(\frac{2b_j+\nu_j+1+\kappa}{\mu_j})}{\Gamma(\frac{\nu_j+1}{\mu_j})}\left[\frac{\Gamma(\frac{\nu_j+1}{\mu_j})}{\Gamma(\frac{\nu_j+2}{\mu_j})}\right]^{2b_j+\kappa} \qquad \theta_j^\kappa = \frac{\Gamma(\frac{2\beta_j+2b_j+\nu_j+1+\kappa}{\mu_j})}{\Gamma(\frac{2b_j+\nu_j+1+\kappa}{\mu_j})}\left[\frac{\Gamma(\frac{\nu_j+1}{\mu_j})}{\Gamma(\frac{\nu_j+2}{\mu_j})}\right]^{2\beta_j} \tag{B5}$$

These non-dimensional values and their derivation are given in greater detail in Seifert (2002) and Seifert and Beheng (2006).

## Appendix C: Notation

$\gamma$ Decay rate in the fragment number generated from ice-ice collisional breakup

$\lambda$ Rate parameter in the generalized $\Gamma$ distribution

$\mu, \nu$ Factors within the shape parameter of the generalized $\Gamma$ distribution

5    $\sigma$ Standard deviation in the raindrop shattering probability distribution function

**ICNC** In-cloud ice crystal number concentration

**INP** Ice-nucleating particle number

$N_{i,pri}$ Primarily nucleated ice crystal number concentration

$N_{i,sec}$ Secondarily produced ice crystal number concentration

10    $N_R$ Raindrop number

$\aleph_{BR}$ Fragment number from ice-ice collisional breakup per collider (e.g., graupel in *ig* or *sg* simulations) number

$\aleph_{DS}$ Fragment number from droplet shattering per large droplet number

$\aleph_{RS}$ Fragment number from rime splintering per milligram of rime

$p_{DS}$ Temperature dependent probability that a freezing raindrop shatters

$P_{freez}$ Probability that a raindrop freezes versus time according to Bigg (1953)

$p_{max}$ Maximum probability that a freezing raindrop shatters, parameter within $p_{DS}$

$\dot{P}$ Precipitation intensity

$P_{tot}$ Accumulated precipitation

$T_{min}$ Minimum temperature for ice-ice collisional breakup to occur

$T_\mu$ Associated temperature for the maximum in the raindrop shattering probability distribution function

$V_R$ Raindrop volume

$\overline{x_R}$ Mean raindrop mass

*Author contributions.* SCS and CH constructed the parameterizations and chose the case study. SCS implemented the parameterizations in COSMO. CB and IZ assisted with model setup and porting, and JC provided campaign data and analysis codes. SCS, CH, and AN analyzed simulation output. All authors reviewed the writing.

*Competing interests.* The authors declare that they have no conflict of interest.

*Acknowledgements.* The comments of two anonymous reviewers have substantially improved our analyses. SCS and CH thank support from the Helmholtz Association through the President's Initiative and Networking Fund (VH-NG-620). SCS and AN acknowledge funding from a NASA Earth and Space Science Fellowship (NNX13AN74H) and NASA MAP grant, as well as the DOE EaSM. JC and the source campaigns for our data were funded by the NERC APPRAISE programme, grant number NE/E01125X/1. All authors wish to thank the
Deutscher Wetterdienst (DWD) for providing the COSMO model code as well as the initial and boundary data. In addition, SCS would like to thank Marco Paukert, Isabelle Reichardt, Tobias Schad, Heike Vogel, and Luke Hande for helpful discussion about the COSMO model during a six-month stay at the Institute of Meteorology and Climate Research in Karlsruhe.

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
