# Peer review of "The effect of secondary ice production parameterization on the simulation of a cold frontal rainband"

_Atmospheric Chemistry and Physics, 2018_

## Referee Comment (RC1) · Anonymous Referee #1 · 23 Jul 2018

The study simulates a frontal rain band over the UK for which extensive radar and insitu observations exist. Two new secondary production parameterizations are introduced to the COSMO model. The modeling is extensive in that it covers 16 sensitivities (+1 control) with 10 perturbations per configuration. These data are then used to compare to observations and assess the impact of the new parameterizations on ice number concentration and precipitation.

I think the study will be publishable after the following comments are addressed.

1/ My main issue is the experimental design (or my interpretation of it). To me, the problem is that the study appears to mix one secondary production process into the control

model and then looks at the impact of the two new processes and modifications to the rime-splintering process as 'the effect of secondary ice production...'. From what i can see in table 2 there is no sensitivity with all secondary processes off. At the same time, the control model is stated as using Seifert and Beheng microphysics. The Seifert and Beheng described in the literature included rime splintering ice production. There are some comments later in the paper that suggest to me that the control does have rime splintering off. Therefore, either i have misinterpreted and some additional description of the control model configuration is needed, or I think we need an additional set of control runs that have no secondary ice production processes included

Until there is a clean control with no secondary ice production processes it is difficult to interpret the statements about the impact of secondary ice production.

2/ The thrust of the paper is to show the impact of the different secondary ice production processes. Given the potential of introducing many unknown parameters into the model I think that it would be really useful for the community to try and identify if certain processes can be ignored. At the moment the paper is pushing us to try to represent more complexity, but it would be advantageous if simplifications could be identified.

For instance, questions that arise as i read through the paper include:

i. what is the relative impact of rime-splintering to droplet shattering to collisional breakup? (something similar was done for primary versus secondary)

ii. Can we ignore any of them or do they interact?

iii. Given the number of unknowns is it possible to write a single parametrization that captures all of the processes with less parameters? (this one would be speculation)

To try and answer i) and ii) the following families of model runs are suggested -some of which you already have.

a) control : no secondary ice production

b) control + rime splintering

c) control + rime splintering + droplet shattering

d) control + rime splintering + droplet shattering + collisional breakup

e) control + droplet shattering

f) control + droplet shattering + collisional breakup

g) control + collisional breakup

h) control + collisional breakup + rime splintering

....

Additional points:

p5 eqn 7. qrim is not defined. Is it the rate of change of ice due to riming?

p6 table2. SY not defined (typo on p5?)

p6, 13. CTRL=Seifert and Beheng - this has rime splintering secondary production as standard? [see main comment above]

p8, 20. Maybe change secondarily-> secondary, primarily -> primary?

p8, 28-29. Is COSMO able to capture this sort of mixing process?

p8, 30-p9,3. 1Ag and 1Ac contain 2 changes. Its difficult to say which change is most important. Including 1An would provide a way to decide the relative importance of the changes.

p10,7: 'There are no heating...' - do you mean because the structures are outside of the traditional temperature range for rime splintering?

p10,8-10. 'Zhu et al....'. I don't know if this can be inferred. I would imagine the melting differences affect the strength of downdrafts and cold pools leading to changes in subsequent convection which would be substantially different to the dynamical coupling due to latent heating in updrafts.

p10,13-14. '...smaller droplets form, diminishing the riming...'. I think making smaller droplets could lead to increase cloud liquid water that would lead to increased riming.

p10,16. Ni,sec - does this include all rime-splintering too? How does it compare to Ni,pri in a control model with no secondary ice production?

p10,27. This is a large domain that includes some of the warm front too?

p10,30. '...filtered out'. These are in-cloud values then. Is it same thresholding for the observations?

p11,2. Control simulation - does this include rime splintering?

p11. figure 4. How many points go into the 10^-3 and lower probability for the observations? If it is less than ∼10 it might be good to ignore them?

p12, 26. 'The control simulation without secondary ice...'. Is this the additional secondary ice processes or all secondary ice including rime splintering? If it is the latter then that needs to be made clear that the control has no secondary ice processes in it (see main point 1 and p11,2, p6,13)

p12,32. How does figure 5 compare to observed accumulations?

p13,3. Please could you also give the domain mean precipitation change?

p13, 8. The red (positive) regions are also correlated with a combination of the location of the front and the orography, where convection may be enhanced.

p14, 4. The structures in Crosier et al are on the scale of ∼5km, whereas these are much bigger ∼50km?

p14, 9-11. 'suggesting that rime splintering is responsible for much of the change in Ptot'. Does this mean that the control did not have rime-splintering?

p14, 14: NCRF not defined?

p14, 34. The variation in the means is now within 10% of the mean of the ensemble of results.

p15, 1-6. Why not use the model to provide the diagnosed rates?

p17. It would be good to see answers to the points raised in 2/ above. It would also be good to state what the overall domain mean change in precipitation is due to secondary ice production processes.

---

## Referee Comment (RC2) · Anonymous Referee #2 · 3 Aug 2018

General Comments:

The authors conduct a numerical modeling study of secondary ice production and its possible effects on surface precipitation, based upon observations of the 3 March 2009 cold frontal passage across Southern England analyzed by Crosier et al. (2014). They address three kinds of secondary ice production: rime-splintering, shattering of freezing raindrops, and collisional ice breakup, with several variations of coefficients that control the magnitude of each process. By enhancing the magnitudes of each process, often to the extent above that represented in past studies, or justified by laboratory experiments, they find (unsurprisingly) that the ice number concentrations can be greatly

increased beyond that expected from primary nucleation. Through some (very rough) comparisons with the observations, they are still unable to replicate the maximum observed ice crystal number concentrations, and yet, estimates of the ice production rate actually exceed those based upon the observations. This disparity suggests that there might be some issues in comparing the observations and simulations, and/or that the simulated cloud dynamics are significantly different than in the observed clouds. The authors find an increase in surface precipitation of ∼20% for the simulations that maximize secondary ice production, and advocate from that result that parameterizations of these processes should be included in large-scale models.

Specific comments:

The authors are tackling a very difficult problem here, and studies like this are important and necessary. However, more care must be taken in what they can and cannot conclude from this study. I have some serious concerns with how the authors conducted some of the analyses, and/or their interpretations. In the manuscript, some very important details are omitted, that make it difficult to understand their interpretations.

Overall, I would like to see this study move forward, but feel that it would be of greater use to emphasize the temperature and dynamical regimes over which each of the secondary ice production processes is dominant, and how those differences assist (or do not) the formation of additional precipitation. That might be a more useful place to start when advocating that some of these processes be included in larger-scale models, as it would help focus case studies of the type of weather phenomena where they could have the most impact.

1. If observations and simulations are compared in this way, particularly when convective elements are contained within the weather phenomenon of interest, then it first must be demonstrated that the clouds and precipitation due to the cold frontal passage in the control case are consistent with that observed, and if not, to state clearly how they differ, and continue compensating for those differences when comparing the observed

and simulated microphysical development. For example, a figure showing simulated radar echo can be produced, and shown alongside Crosier's Fig. 3, to understand how the general structure might differ. The timing is also important: if observations over a given time are averaged and compared with the simulations, any issues in doing so must be known. A small paragraph summarizing the dynamics of the observed clouds, based on the analysis of Crosier et al., would also be helpful in "setting the stage" for the reader, regarding the types of clouds (strengths of updraft speeds measured by aircraft, cloud top temperatures) being considered here.

2. The routes from ice to precipitation discussed in the introduction, shown in Fig. 1, and later discussed with respect to effects of the secondary ice upon precipitation, are not inclusive of a major route from ice crystals to precipitation: enhanced rimed ice/rimed snow/graupel/frozen raindrop formation that can melt to become surface precipitation. In Fig. 1, it is somewhat suggested by (3), but the arrow isn't drawn as leading to acceleration of precipitation like (2). Such an analysis of that route to precipitation is completely omitted in the manuscript. Why? Even if no graupel were observed, the Crosier et al. paper discussed the importance of rimed snow, and noted that the aircraft did not sample the stronger convection where graupel might have resided. The authors only discuss in this study the possible effects on the Bergeron process leading to precipitation, but that would be more important in the stratiform precipitation regions, and not as much in the convective regions of the cold front band, where the heaviest precipitation will fall. I would think that the precipitation enhancement seen in Fig. 5 is due to rimed particles, not from an enhanced Bergeron process.

3. The implementation of the secondary ice parameterizations in the two-moment Seifert and Beheng scheme are confusing.

a. Why is the second moment not taken advantage of, here? Everything seems to depend upon mass. For example, rime-splintering appears to have a constraint of rimed mass, but for a two-moment scheme, the Cotton et al. (1986) second formulation that uses the number of fragments per number of 25 um diameter drops accreted would

be a better prediction. The lab studies have shown that if the rimed drops don't achieve this size, they won't splinter. As implemented here, there is no drop size dependence, so splintering might be greatly overestimated, and some commentary needs to be given in regards to that limitation.

b. Along similar lines, what is the justification for the experiments using broader temperature ranges and/or increased fragment numbers for rime-splintering? That process has been studied much more in the laboratory than others (Hallet and Mossop, and Sauners & Hosseini, AR, 2001). The results here seem to rely on the expansion of this process to a broader area of temperature than appears justified by the laboratory work. The latter study also looks at the importance of fall speed, where graupel is more favored for greater splinter production. Since the simulation has little/no graupel here, then it would seem to imply using smaller splintering rates is appropriate.

c. Also, there is no mention of the recent work on ice-ice collisions, and its parameterization, by Phillips et al. (Phillips et al., JAS, 2017 and 2019), or for shattering of freezing drops (Phillips et al., JAS, 2018). How do their parameterizations compare to those used here, and how might that influence differences in the effects upon precipitation?

4. It is stated that Crosier et al. noted fall streaks at cloud top in the radar measurements. This would seem to imply a seeding mechanism of ice from above that could also have fallen to the observation level of the aircraft, unless this has somehow been ruled out?

5. I would contend that most "larger-scale models" do not have two-moment microphysical schemes, so that the suggestion at lines 21-24 on page 10 are not practical.

6. Comparison of observations and modeling: unless the authors can justify that the simulated movement of the rainband, and its dynamical nature, was very similar to that observed, it would make the comparisons shown in Fig. 4 not very useful. (If the comparison is not "fair", it could even be the case that the model IS producing

sufficient secondary ice, for example, if the dynamical/thermodynamical conditions of the simulated clouds in that region and over that time are different than observed!) The description of the analysis for panels b and c on page 12 is helpful, but the reader cannot see what is being compared (which is why some comparison of the observed radar evolution to the simulated one is needed early on in the paper.) The temperature ranges also need to specified in Fig. 4, too.

7. Ice production rates: it needs to be stated more clearly how these were derived, from both the observations and the modeling results. Right now, it is unconvincing that this comparison is valid. Also, it should be stated somewhere that the CIP-15 observations were corrected using an algorithm designed to remove shattering artifacts, but some still likely remain because anti-shattering tips were not used on the instrument, as stated by Crosier et al.

8. Qualifiers/limitations: need to be stated clearly throughout the paper. For example:

a. The model uses the primary ice nucleation parameterization of Phillips et al. (2008). Since INP measurements were not collected, it is unknown if this is an accurate rep-resentation, or not, and this might greatly affect the ratios of secondary to primary nucleated ice, including the possible importance of secondary ice to precipitation.

b. To show an appreciable affect on surface precipitation (20% increase), rime-splintering had to be increased over that typically depicted in models based on the laboratory measurements (e.g. Cotton et al. 1986 parameterization).

c. Reasons for why the other two secondary ice processes might be less important here: (i) minimal graupel, which is important for ice-ice collisional breakup; (ii) limited number of raindrops? (Not sure what else would have limited that process here, but it would be good to know.) Also, it should be explained that the Crosier et al. study noted limited graupel in their observations, and I don't think they found any evidence of shattered frozen raindrops, but they clearly state that the former could have been limited by the inability of the aircraft to fly in the more convective regions.

d. The enhancement of the updrafts and precipitation, and downdrafts, mentioned on page 14 needs to be backed up with some evidence.

e. Some discussion of model resolution effects should be included, for both properly simulating the microphysics as well as the dynamics.

Technical Corrections:

1. Table 2 and Fig 2 are confusing, since most of these runs are not discussed in the paper. I would suggest only showing those that are discussed here, and just noting somewhere (if important) that other variations did not show much change in the results. Also, if the text could explain the naming convection for the different simulations used here, or new labels that are more intuitive to the particular change in a given simulation used, that would be much easier for the reader to interpret them.

If the major issues in this review are addressed, I suspect that much of the wording will be revised, and thus I will refrain from noting any suggestions at this time.

---

## Author Comment (AC1) · 18 Oct 2018

**Reviewer 1 Responses**

The study simulates a frontal rain band over the UK for which extensive radar and in-situ observations exist. Two new secondary production parameterizations are introduced to the COSMO model. The modeling is extensive in that it covers 16 sensitivities (+1 control) with 10 perturbations per configuration. These data are then used to compare to observations and assess the impact of the new parameterizations on ice number concentration and precipitation.

I think the study will be publishable after the following comments are addressed.

Thank you for your thorough reading and suggestions to improve it.

1/ My main issue is the experimental design (or my interpretation of it). To me, the problem is that the study appears to mix one secondary production process into the control model and then looks at the impact of the two new processes and modifications to the rime splintering process as 'the effect of secondary ice production...'. From what I can see in Table 2 there is no sensitivity with all secondary ice processes off. At the same time, the control model is stated as using Seifert and Beheng microphysics. The Seifert and Beheng described in the literature included rime splintering ice production. There are some comments later in the paper that suggest to me that the control does have rime splintering off. Therefore, either I have misinterpreted and some additional description of the control model configuration is needed, or I think we need an additional set of control runs that have no secondary ice production processes included.

Until there is a clean control with no secondary ice production processes it is difficult to interpret the statements about the impact of secondary ice production.

While we had not included the control run (CTRL) in the original version of Table 2, it was indeed done without any secondary ice production processes active; the default rime splintering of the Seifert and Beheng scheme was turned off. Thank you for pointing out that this was not explicit. In Section 3, we have rewritten: *A control simulation [is also run] in which all secondary ice production processes, including the default rime splintering in SB06, are turned off (denoted `CTRL' throughout).* We have also addressed some of the discussion of the CTRL versus non-control simulations below.

2/ The thrust of the paper is to show the impact of the different secondary ice production processes. Given the potential of introducing many unknown parameters into the model I think that it would be really useful for the community to try and identify if certain processes can be ignored. At the moment, the paper is pushing us to try to represent more complexity, but it would be advantageous if simplifications could be identified.

For instance, question that arise as I read through the paper include:

    i.       What is the relative impact of rime splintering to droplet shattering to collisional breakup? (Something similar was done for primary versus secondary)

    ii.      Can we ignore any of them or do they interact?

    iii.    Given the number of unknowns is it possible to write a single parameterization that captures all of the processes with less parameters? (this one would be speculation)

To try and answer i) and ii) the following families of model runs are suggested – some of which you already have.

    a)   Control: no secondary ice production (CTRL)
    b)   Control + rime splintering (RS1, RS2)
    c)   Control + rime splintering + droplet shattering (RS2+DS2)

d) Control + rime splintering + droplet shattering + collisional breakup (ALL)
e) Control + droplet shattering (DS1, DS2)
f) Control + droplet shattering + collisional breakup (DS2+BR2ig)
g) Control + collisional breakup (BR1ig, BR2ig, BR2sg)
h) Control + collisional breakup + rime splintering (RS2+BR2ig)

At both your and the other reviewer's suggestion, we have adjusted the study format. Some of the parameters used for rime splintering or droplet shattering were overly generous and results were not shown from several of the simulations in the first version of Table 2. We had also done no simulations with a single process, but rather with rime splintering *and* either drop shattering *or* breakup because part of our intent was to see feedbacks between these processes. In particular, we had been hoping to confirm or deny the existence of a kind of cascade effect (as proposed by Lawson et al. 2015) in which a few ice crystals formed from an initial droplet shatter or ice-ice collision then kick start rime splintering. To address the relative importance of the processes (your questions i and ii) for various thermodynamic regimes (your question iii in part and noted by the other reviewer), we have reorganized the simulations to be the following (summarized in a new Figure 2 and denoted above):

**Rime splintering**

| | | |
|---|---|---|
| **RS1:** | $\aleph_{RS} = 300,$ | $w_{RS} = TR$ |
| **RS2:** | $\aleph_{RS} = 300,$ | $w_{RS} = UNI$ |

**Ice-ice collisional breakup**

| | | | |
|---|---|---|---|
| **BR1ig:** | *graupel_breakup_ice* | | |
| | $F_{BR} = 180,$ | $T_{min} = 256,$ | $\gamma = 3$ |
| **BR2ig:** | *graupel_breakup_ice* | | |
| | $F_{BR} = 360,$ | $T_{min} = 249,$ | $\gamma = 5$ |
| **BR2sg:** | *graupel_breakup_snow* | | |
| | $F_{BR} = 360,$ | $T_{min} = 249,$ | $\gamma = 5$ |

**Droplet shattering**

| | | | |
|---|---|---|---|
| **DS1:** | $\aleph_{DS} = 2,$ | $p_{max} = 5\%,$ | $\sigma = 3$ |
| **DS2:** | $\aleph_{DS} = 10,$ | $p_{max} = 10\%,$ | $\sigma = 5$ |

**Combinations**

| | | | |
|---|---|---|---|
| **RS2 + BR2ig:** | $\aleph_{RS} = 300,$ | $w_{RS} = UNI$ | |
| | *graupel_breakup_ice* | | |
| | $F_{BR} = 360,$ | $T_{min} = 249,$ | $\gamma = 5$ |
| **DS2 + BR2ig:** | $\aleph_{DS} = 10,$ | $p_{max} = 10\%,$ | $\sigma = 5$ |
| | *graupel_breakup_ice* | | |
| | $F_{BR} = 360,$ | $T_{min} = 249,$ | $\gamma = 5$ |
| **RS2 + DS2:** | $\aleph_{RS} = 300,$ | $w_{RS} = UNI$ | |
| | $\aleph_{DS} = 10,$ | $p_{max} = 10\%,$ | $\sigma = 5$ |

**Control**

| | | |
|---|---|---|
| **CTRL:** | $\aleph_* = 0$ | $F_{BR} = 0$ |

| | | | |
|---|---|---|---|
| **ALL:** | $\aleph_{RS} = 300,$ | $w_{RS} = UNI$ | |
| | $F_{BR} = 360,$ | $T_{min} = 249,$ | $\gamma = 5$ |
| | *graupel_breakup_** | | |
| | $\aleph_{DS} = 10,$ | $p_{max} = 10\%,$ | $\sigma = 7$ |

[Figure]

*Figure 2 Fragment numbers, weightings, and probabilities from the secondary ice production parameterizations. In panel a, we show $N_{BR}$ from both ice-ice collisional breakup simulations (BR1 and BR2) as well as the triangular and uniform $w_{RS}$ (T). In panel b, we show $p_{DS}$ from both droplet shattering simulations (DS1 and DS2) and $w_{RS}$ once again.*

To address your questions about relative importance and interaction, we have added a Figure 6 which compares the contributions from rime splintering, droplet shattering, collisional breakup, and all in combination in a 2 x 2 panel, as the old Figure 4 did for the primarily and secondarily formed ice crystal numbers.

[Figure]

*Figure 6 Maps to compare $N_{i,sec}$ between 18:00 UTC and 18:30 UTC from (a) rime splintering (RS1 at 3 km) (b) frozen drop shattering (DS1 at 4.5 km) (c) collisional breakup between snow and graupel (BR2sg at 1.5 km) and (d) all secondary processes occurring simultaneously (ALL at 4.5 km). Note the different logarithmic colorbars for each panel.*

We have also added discussion of this to Section 4.2:

*Next, we consider the relative ice crystal number concentrations produced by different processes in Figure 6. The largest $N_{i,sec}$ magnitudes, up to 1000 $L^{-1}$ over the half hour, come from the RS1 and ALL simulations. These are followed by about 10 $L^{-1}$ (half hour)$^{-1}$ generation rates from frozen droplet shattering and 0.1 $L^{-1}$ (half hour)$^{-1}$ from collision breakup of snow and graupel. There is also an altitudinal hierarchy. Contributions from droplet shattering are largest at the highest altitudes of 4.5 km where raindrop number concentrations are still relatively high and the temperature (T in [237 K, 262 K] with a median of 249 K) is cold enough for non-negligible shattering probability. The rime splintering contribution is next at an altitude of 3 km, and the breakup is largest at a lower altitude of*

*1.5 km because the graupel mixing ratio is highest here. If graupel were present at higher altitudes, $N_{i,sec}$ from breakup could increase significantly, as both the snow mixing ratio and fragment number parameter increase at colder temperatures.*

*Finally, we have included fields of graupel and snow mixing ratios and rain drop number concentrations in the supplemental information, as well as a new section on dynamical intercomparisons – of wind speed, updrafts, and radar reflectivites – to make the discussion more thorough.*

Additional points:

p. 5 eqn. 7. qrim is not defined. Is it the rate of change of ice due to riming?
$q_{rim}$ is the rime mixing ratio. We add this to the description proceeding equation 7.

p. 6 table 2. SY not defined (typo on p. 5?)
We have eliminated the simulations that employed this symmetric temperature weighting for rime splintering (SY was for symmetric.)

p. 6, 13. CTRL = Seifert and Beheng – this has rime splintering secondary production as standard? [see main comment above]
You are right that in the default set-up Seifert and Beheng includes rime splintering. But we turn it off for our CTRL simulation and have made this clearer throughout the rewrite.

p. 8, 20. Maybe change secondarily -> secondary, primarily -> primary
We would prefer to leave the terminology as "secondarily-produced ICNC" / "primarily-nucleated ICNC" or "ICNC from secondary production" / "ICNC from primary nucleation".

p. 8, 28-29. Is COSMO able to capture this sort of mixing process?
We went ahead and removed this comment, as there is not extensive discussion of how COSMO represents vertical mixing in the model description.

p. 8, 30-p. 9, 3. 1Ag and 1Ac contain 2 changes. It's difficult to say which change is most important. Including 1An would provide a way to decide the relative importance of the changes.
Given the relatively large amount of observational evidence that suggests that the fragment number per milligram of rime should be on the order of $10^8$, we have limited the rime splintering simulations to two with the standard triangular temperature weighting and a slightly extended one. So we will not discuss different fragment numbers.

p. 10, 7. 'There are no heating...' – do you mean because the structures are outside of the traditional temperature range for rime splintering?
We have moved this paragraph to the discussion section. The idea was that because rime splintering is a mechanical process, simply the shedding of fragile protuberances from rime, there is no latent heat release or consumption when it occurs. We have clarified: *"As mechanical processes, rime splintering and collisional breakup do not have direct latent heating effects."*

p. 10, 8-10. 'Zhu et al...' I don't know if this can be inferred. I would imagine the melting differences affect the strength of downdrafts and cold pools leading to changes in subsequent convection which would be substantially different to the dynamical coupling due to latent heating in updrafts.
*Yes, thank you. A more relevant study is that of Willison et al. 2013 The importance of resolving mesoscale latent heating in the North Atlantic Storm Track in which they discuss the latent heating effects on mid-latitude cyclogenesis and their spatial resolution dependence. We write:*
*Additional latent heating aloft can intensify the upper-level anticyclonic potential vorticity (PV) [Willison et al. 2013]. This PV generation may suppress further cyclogenesis by disconnecting the*

*developing system from surface potential temperature anomalies. On the other hand, additional cyclonic diabatic PV may slow system progression and maintain favorable levels of shear.*

p. 10, 13-14. '...smaller droplets form, diminishing the riming...'. I think making smaller droplets could lead to increase cloud liquid water that would lead to increased riming.
Yes, this is true, but we were thinking of a case of fixed cloud LWC. In this case, more and smaller droplets should decrease the collisional efficiency between these and ice hydrometeors. We have made the fixed cloud LWC assumption explicit.

p.10, 16. Ni,sec – does this include all rime-splintering too? How does it compare to Ni,pri in a control model with no secondary ice production?
In the updated figure, $N_{i,sec}$ is indeed the secondary ice produced from rime splintering. Instead of showing its ratio relative to the primarily nucleated ice $N_{i,pri}$ we have chosen to show the absolute $N_{i,pri}$. This removes some numerical concerns (where one or the other tendency was absent and division created unrealistic values). Although we do not include it in the updated manuscript, here is an $N_{i,pri}$ field from the CTRL simulation:

[Figure]

It has similar structure but somewhat smaller magnitude than the same field shown for the RS1 and RS2 fields in Figure 5.

p. 10, 27. This is a large domain that includes some of the warm front too?
Yes, this is true. It is the subdomain shown in black boxes in the new Figures 5 and 6, so an area that should also contain some of the warm front. We mention this in reference to some of the underestimation: *A final contribution to these too low ICNCs may be inclusion of parts of the warm front in the subdomain of analysis.*

p. 10, 30. '...filtered out'. These are in-cloud values then. Is it same thresholding for the observations?
The measurement accuracy does not extend to this low level, so we did not filter the observations in the same manner.

p. 11, 2. Control simulation – does this include rime splintering?
No, there is no secondary ice production in the control simulation.

p. 11, figure 4. How many points go into the $10^{-3}$ and lower probability for the observations? If it less than ~10 it might be good to ignore them?
We would prefer to keep the low-probability tail in Figure 7b to indicate the high degree of skewedness in the distribution. These instances, even if very few, are the ones most likely to reflect secondary ice production since $N_{ice}$ is so high.

p. 12, 26. 'The control simulation without secondary ice...'. Is this the additional secondary ice processes or all secondary ice including rime splintering? If it is the latter then that needs to be made clear that the control has no secondary ice processes in it (see main point, p. 11,2, p. 6,13).

There is no secondary ice production in the control simulation. To remove confusion we delete "without secondary ice" here. In Section 3 on the Simulations, we write out: *"a control simulation in which all secondary ice production processes are turned off, including the default rime splintering in SB06 (denoted CTRL throughout)."*

p. 12, 32. How does figure 5 compare to observed accumulations?

The UK NIMROD radar data only offers rainfall rates. If we look instead at the CFARR ground site measurements of precipitation rate (Crosier et al. 2014 Figure 4b) we can very roughly integrate it. Say from 18:00 to 20:30 UTC there is a rate of 1.5 mm h$^{-1}$ (= 4.5 mm) and another 1 mm h$^{-1}$ from 21:00 to 22:00 (= 1 mm). If the rainband passage happens over a half hour with 60 mm h$^{-1}$ intensity then we have a total accumulation of 4.5 + 1 + (0.5)(60) = 35.5 mm in the regions that saw the maximum precipitation intensity. This is not far off from the simulated values.

p. 13, 3. Please could you also give the domain mean precipitation change?

Yes, this is a good idea, thank you. We have written *"The sum of the deviations over the whole subdomain is an additional 23.9 m of precipitation for the RS1 simulation, 25.6 m for RS2, 16.9 m for DS1, and 16.6 m for ALL."*

p. 13, 8. The red (positive) regions are also correlated with a combination of the location of the front and the orography, where convection may be enhanced.

Indeed. In reference to the location of the front, we write that *"Banding [in the P$_{tot}$ deviations] reflects convective structure: vertical motion is strongest in the rainband leading edge, but also preceded and proceeded by downdrafts."* We have shown some of this dynamical structure in the new Figure 3c.

We had not considered the effect of orography; thank you for this suggestion. It could be that the particularly high P$_{tot}$ seen in Figure 8a around 50.5°N and 4°W is due to orographic lifting by Dartmoor. We add the following description of topography to Section 4.3:

*Orography in this region also has an impact. The spot of particularly large P$_{tot}$ around 50.5°N and 4°W corresponds to the Dartmoor with a maximum elevation of 621 m. Slightly elevated P$_{tot}$ is also present over the Exmoor and Bodmin Moor at (51°N, 3.5°W) and (50.5°N, 4.5°W).*

p. 14, 4. The structures in Crosier et al are on the scale of ~5 km, whereas these are much bigger ~50 km?

Yes, you are right. A direct comparison of banded structure in the differential reflectivity and accumulated precipitation fields does not make total sense, so we have removed this particular comment. In general, however, we see "broadening behavior" in the simulations relative to the observations. For example, when we compare observed versus simulated radar reflectivity, we see that Z$_{DH}$ values have a too-low magnitude over a too-great extent:

[Figure]

*Figure 4 Model-measurement intercomparison of range-height indicator scans of radar reflectivity $Z_{DH}$ along the 255 degree radial out from CFARR. CAMRa Doppler radar measurements are shown in panel a for the scan taken between 19:22:07 and 19:23:07 UTC, and modelare shown from the CTRL simulation at 19:00:00 UTC, both in $dBZ_h$.*

Or in a qualitatively similar manner, we see too-low precipitation intensity magnitudes over too-great a time period in Figure 9.

p. 14, 9-11. 'suggesting that rime splintering is responsible for much of the change in $P_{tot}$.' Does this mean that the control did not have rime-splintering?
Yes, there is no secondary ice production in the control simulation. In Section 3 on the Simulations, we write out: *"a control simulation in which all secondary ice production processes are turned off, including the default rime splintering in SB06 (denoted CTRL throughout)."*

p. 14, 14. NCRF not defined?
Yes, thank you for pointing this out. We have added *narrow cold frontal rainbands (NCFRs).*

p. 14, 34. The variation in the means is now within 10% of the mean of the ensemble of results.
The differences in precipitation intensity vary less between simulations and deviate less from the CTRL simulation than those in precipitation accumulation. You are right that the difference in precipitation intensity from one simulation to another is not statistically significant. To the discussion in Section 4.3, we make explicit that *"In neither case does the evolution of the mean precipitation intensity for different simulations vary significantly from one to the next."*

p. 15, 1-6. Why not use the model to provide the diagnosed rates?
Yes, this is a good point. We opted to drop this analysis since it had significant numerical noise. Instead we have replaced these with temporal evolutions of the $N_{ice}$ profile over time (in part to investigate the impact of seeding):

[Figure]

*Figure S3 Temporal evolution of $N_{ice}$ (panels a, b, c) and $N_{pri}$ (panels d, e, f) profiles in the RS1 simulation from three, randomly-sampled latitude / longitude locations in the vicinity of CFARR. Eight profiles are shown for each location, one for each half hour from 18:00 UTC to 21:30 UTC with the darker colors representing earlier times and the lighter ones later times.*

p. 17. It would be good to see answers to the points raised in 2/ above. It would also be good to state what the overall domain mean change in precipitation is due to secondary production processes.

We have added to the enumerated conclusions to address relative importance and parameterizability of these processes. To the first conclusion on ice production rates from primary nucleation versus secondary production, we build off the new Figure 6 and add:

"*In this case, we saw that rime splintering was the most important process in line with the conclusions of Crosier et al. 2014; however, underestimation of vertical velocities in the cold front also led to underestimation in simulated radar reflectivity relative to observations. If this $Z_{DH}$ difference was caused by additional graupel at higher altitudes, contributions from collisional breakup could have been much higher than the 0.1 $L^{-1}$ per half hour found here. A low bias in updrafts also generates fewer raindrops at altitude and limits the contribution from frozen droplet shattering (in this case to an intermediate production rate of 10 $L^{-1}$ per half hour).*"

In regard to how the processes can best be parameterized we refer to the supplemental figure that shows large hydrometeor number concentrations or mixing ratios. There is a strong relation, for example, in the structure of the $N_{i,sec}$ from collisional breakup and the graupel mixing ratio. In the conclusions, we reiterate that: "*Underestimations stem in part from low biases in the updraft velocity. If the vertical velocities can be brought into agreement with observations, then criteria in these values as well as temperature could be used together to parameterize secondary production in appropriate thermodynamic zones. For two-moment schemes, graupel, snow, and raindrop criteria could be implemented for these processes.*"

[Figure]

*Figure S5 Graupel mixing ratio (a), snow mixing ratio (b), large-scale graupel quantity (c), and rain drop number concentration (d) in the simulation domain at 18:00 UTC for the RS2 simulation.*

In regard to the second point, as mentioned in the response for p. 13, 3 above, we have written "*The sum of [$P_{tot}$] deviations over the whole subdomain is an additional 23.9 m of precipitation for the RS1 simulation, 25.6 m for RS2, 16.9 m for DS1, and 16.6 m for ALL.*"

---

## Author Comment (AC2) · 18 Oct 2018

**Reviewer 2 Responses**

**General comments**:

The authors conduct a numerical modeling study of secondary ice production and its possible effects on surface precipitation, based upon observations of the 3 March 2009 cold frontal passage across Southern England analyzed by Crosier et al. (2014). They address three kinds of secondary ice production: rime splintering, shattering of freezing raindrops, and collisional ice breakup, with several variations of coefficients that control the magnitude of each process. By enhancing the magnitudes of each process above that represented in past studies, or justified by laboratory experiments, they find (unsurprisingly) that the ice number concentrations can be greatly increased beyond that expected from primary nucleation. Through some (very rough) comparisons with the observations, they are still unable to replicate the maximum observed ice crystal number concentrations, and yet, estimates of the ice production rate actually exceed those based upon the observations. This disparity suggests that there might be some issues in comparing the observations and simulations, and/or that the simulated cloud dynamics are significantly different than in the observed clouds. The authors find an increase in surface precipitation of ~20% for the simulations that maximize secondary ice production, and advocate from that result that parameterizations of these processes should be included in large-scale models.

Thank you for your thorough reading of the work and suggestions to make it more rigorous. At both reviewers' behest, we have spent significant time to redo the simulations, adjust the structure of the study, and visualize new data from APPRAISE.

For the simulations, some of the parameter values that were chosen, particularly for rime splintering and droplet shattering were overly generous as you state. This was done, in part, to see if we were able to bring model and measured values into agreement with only modification to the secondary ice microphysics. Again as you state, we did not have agreement even at these extreme values, and so we have retained more conservative parameter values (sticking to, for example, 300 fragments (mg rime)$^{-1}$ for rime splintering) and modified the simulations as shown in the new Table 2 at the end of these responses.

Then we have added a section dedicated to reviewing the dynamical environment prior to any discussion of microphysical or precipitation (see responses to Comments 1, 6, and 8). This includes comparisons of radar reflectivity, updraft velocity, and surface wind speed.

**Specific comments**:

The authors are tackling a very difficult problem here, and studies like this are important and necessary. However, more care must be taken in what they can and cannot conclude from this study. I have some serious concerns with how the authors conducted some of the analyses, and/or their interpretations. In the manuscript, some very important details are omitted, that make it difficult to understand their interpretations.

Overall, I would like to see this study move forward, but feel that it would be of greater use to emphasize the temperature and dynamical regimes over which each of the secondary ice production processes is dominant, and how those differences assist (or do not) the formation of additional precipitation). That might be a more useful place to start when advocating that some of these processes be included in larger-scale models, as it would help focus case studies of the type of weather phenomena where they could have the most impact.

1. If observations and simulations are compared in this way, particularly when convective elements are contained within the weather phenomenon of interest, then it first must be demonstrated that the clouds and precipitation due to the cold frontal passage in the control case are consistent with that observed, and if not, to state clearly how they differ, and continue compensating for those differences when comparing the observed and simulated microphysical development. For example, a figure showing simulated radar echo can be produced, and shown alongside Crosier's Fig. 3 to understand how the general structure might differ. The timing is also important: if observations over a given time are averaged and compared with the simulations, any issues in doing so must be known. A small paragraph summarizing the dynamics of the observed clouds, based on the analysis of Crosier et al., would also be helpful in "setting the stage" for the reader, regarding the types of clouds (strengths of updraft speeds measured by aircraft, cloud top temperatures) being considered here.

   Thank you for pointing out the need for these kinds of dynamical comparisons. We have compared modeled and observed radar reflectivity as you suggest. The comparison cannot be exact, as the field shown in Crosier et al. 2014 (and the CAMRa data in general) have much higher resolution along a radial whereas the model output exist only on the 2.8 km grid. In addition, the range-height indicator (RHI) of Crosier et al. 2014 comes from the CAMRa scan between 192207 and 192307 UTC, whereas we only have model output every 30 minutes.

   Nevertheless, we calculate the latitude-longitude pairs along the 255° radial RHI in Fig 5b and identify the modeled lat-lon that minimize the Euclidean distance between these exact values and the model's spatial discretization. From here to generate an RHI, we iterate over a 1000 x 500 grid of distances and heights (relative to the Chilbolton Facility for Atmospheric and Radio Research) and do a bilinear interpolation over the nearest modeled lat-lons and 29 model levels to generate the radar reflectivities that are shown in the new Figure 4:

[Figure]

*Figure 4 Model-measurement intercomparison of range-height indicator scans of radar reflectivity $Z_{DH}$ along the 255 degree radial out from CFARR. CAMRa Doppler radar measurements are shown in panel a for the scan taken between 19:22:07 and 19:23:07 UTC, and modeled values are shown from the CTRL simulation at 19:00:00 UTC, both in dBz$_h$.*

We use the same methodology to calculate an RHI-type plot of the modeled updraft velocity; however this comparison includes an extra degree of inexactitude, as even the "observed field" is actually derived "using Doppler velocity measurements from RHIs by assuming mass-weighted flow continuity" [Crosier et al. 2014, Chapman and Browning 1998]. We have also compared modeled and observed (from the CFARR ground site) wind speeds and shown panels of modeled updraft velocity at two different altitudes during rain band passage in the new Figure 3:

[Figure]

Figure 5 Different model dynamical fields. In panel a, we show the updraft velocity 150 km along the 255 degree radial from CFARR at 51.1N, 1.4W with values from the CTRL simulation at 19:00 UTC. Discontinuities are due to the minimization of Euclidean distance or interpolation aspects of an algorithm to approximate the radial from the model discretization. Surface wind speeds from 17:00 to 22:00 UTC from the CFARR three cup anemometer and our CTRL simulation are shown in panel b. Panels c and d show the modeled vertical velocities from the CTRL simulation at 1 and 7 km respectively at 18:00 UTC as the rainband began to pass over the UK.

We describe these methods and comment on their impact on microphysical comparisons in a new Section 4.1 on the *Dynamic environment*:

**4.1 Dynamic environment**

[revised manuscript text omitted]

2. The routes from ice to precipitation discussed in the introduction, shown in Fig. 1, and later discussed with respect to effects of the secondary ice upon precipitation, are not inclusive of a major route from ice crystals to precipitation: enhanced rimed ice / rimed snow / graupel / frozen raindrop formation that can melt to become surface precipitation. In Fig. 1, it is somewhat suggested by (3), but the arrow isn't drawn as leading to acceleration of precipitation like (2). Such an analysis of that route to precipitation is completely omitted in the manuscript. Why?

   Thank you for pointing out this missing mechanism. Figure 1 focused on showing how secondary ice production (solely) impacts precipitation via increased ice crystal number concentrations, so we had not included impacts of just riming (only rime splintering). But we agree that a more complete analysis cannot consider secondary ice processes in isolation, and we have added enhancement of precipitation by rimed hydrometeors to the schematic. To the discussion of cloud ice-precipitation linkages, we add the following:

   *Efficient riming at mixed-phase temperatures may also simply generate larger hydrometeors that sediment more quickly, particularly in convective regions with a high degree of mixing.*

   Even if no graupel were observed, the Crosier et al. paper discussed the importance of rimed snow, and noted that the aircraft did not sample the stronger convection where graupel might have resided. The authors only discuss in this study the possible effects on the Bergeron process leading to precipitation, but that would be more important in the stratiform precipitation regions, and not as

much in the convective regions of the cold front band, where the heaviest precipitation will fall. I would think that the precipitation enhancement see in Fig. 5 is due to rimed particles, not from an enhanced Bergeron process.

Elsewhere in the introduction, we note that *in cases of ice-initiated precipitation, the requisite crystal growth can occur **via riming** or the Bergeron process.* In the discussion of ice-precipitation linkages, we note that *this pathway [of small crystal formation depleting supersaturation until the Bergeron process initiates] should be more important for stratiform precipitation, given the narrow range of requisite ambient vapor pressures: indeed for an integral ice radius of 100 um cm$^{-3}$, the updraft must be less than about a 1 m s$^{-1}$ for the Bergeron process to occur [Korolev 2007].* To the discussion of impacts on precipitation, we add that *this amplification may be due in part to more riming in the ascending regions, which feeds into precipitation both directly as rimed particles sediment and melt (Fig. 1(3)) and indirectly as they splinter and generate more rimable particles (Fig. 1(4)). Crosier et al. 2014 note that higher values of differential reflectivity around cloud top could be due to high concentrations of rime particles.*

3. The implementation of the secondary ice parameterizations in the two-moment Seifert and Beheng scheme are confusing.

a. Why is the second moment not taken advantage of here? Everything seems to depend upon mass.

We are indeed taking advantage of the second moment outside of the fragment number parameters. Within the ice-ice collisional breakup formulation for example, the $D_j$ and $D_k$ in Equation 6 are the particle diameters associated with particle mass through a power law as detailed in Appendix B. Within the droplet shattering formulation, the tendency of freezing droplets in time is a function of the mean mass per raindrop as detailed in Appendix A.

The importance of the second moment extends beyond a collision or freezing tendency to the fragment number parameters, and you are right that this dependency has not yet been incorporated. There is not enough consensus of laboratory and in-situ measurements to support one fragment number function in our opinion. We expand the section on the frozen droplet shattering parameterization (Section 2.1) and add to our comment in the earlier version that *future studies should add dependency on droplet size to the ejected fragment number*:

*Recent droplet levitation experiments and high speed video are elucidating the exact physics behind the shattering of droplets as they freeze [Leisner et al. 2014, Wildeman et al. 2017]. Droplet shattering has been previously parameterized statistically in a bin microphysics scheme with the fragment number as a function of drop diameter to the fourth power, using data from the Ice in Clouds Experiment - Tropical (ICE-T) campaign [Lawson et al. 2015, Lawson et al. 2017]. But while measurements continue to confirm a strong dependence of fragment number on droplet size, even recent studies could not confirm this fourth-power dependence [e.g., Lauber et al. 2018]. The laboratory studies of Lauber et al. 2018 in particular add important quantitative results to existing secondary ice measurements but are taken at two droplet sizes (83 and 310 um) so that it remains difficult to rigorously formulate fragment number.*

To the section on the ice-ice collisional breakup parameterization (Section 2.2), we also add discussion of other parameterizations:

*Vardiman first parameterized ice-ice collisional breakup using fragment generation functions based on the momentum exchange between two particles upon impact and leading coefficients dependent upon crystal type [Vardiman78]. More recently, Yano and Phillips 2011 and Yano et al. 2016 constructed a dynamical system-type models that tracks only ice crystal, small graupel, and large graupel number densities and illustrated the ability of ice-ice collisions to generate huge ice crystal enhancements in the absence of vapor limitation. Recently a more complete parameterization has used an exponential formulation with the initial kinetic energy of two particles, their temperature- and humidity-dependent collision type, and asperity fragility coefficients [Phillips 2017a, Phillips 2017b].*

*We choose to focus on temperature dependence in a more straightforward, if less physically rigorous, product of fragment number and hydrometeor collision tendency.*

For example, rime-splintering appears to have a constraint of rimed mass, but for a two-moment scheme, the Cotton et al. (1986) second formulation that uses the number of fragments per number of 25 um diameter drops accreted would be a better prediction. The lab studies have shown that if the rimed drops don't achieve this size they won't splinter. As implemented here, there is no drop size dependence, so splintering might be greatly overestimated, and some commentary needs to be given in regards to that limitation.

*In connection with the previous comment, this statement is a bit confusing, as it advocates for use of the first moment* rather *than the second moment. Nevertheless, we had made the rime splintering parameterization active only for rain drops. At the end of Section 2.3, we had stated: We limit rime splintering to occur only after collisions between raindrops and ice crystals, graupel, or hail. However, raindrops in the Seifert and Beheng scheme have a radius of 40 um, and as you state, the radius of onset for rime splintering is lower, around 12 um. So we have gone back and implemented rime splintering with cloud droplets as well, defining a 12 um threshold radius for these cases. We adjust the Section 2.3 description:*

*We also limit rime splintering to occur only after collisions between cloud droplets of diameter greater than 25 um or raindrops (r > 40 um) and ice crystals, graupel, hail, or snow [e.g., Phillips et al. 2001, Connolly et al. 2006b].*

b.  Along similar lines, what is the justification for the experiments using broader temperature ranges and/or increased fragment numbers for rime splintering? That process has been studied much more in the laboratory than others (Hallet and Mossop and Saunders & Hosseini, AR, 2001). The results here seem to rely on the expansion of this process to a broader area of temperature than appears justified by the laboratory work. The latter study also looks at the importance of fall speed, where graupel is more favored for greater splinter production. Since the simulation has little / no graupel here, then it would seem to imply using smaller splintering rates is appropriate.

Some justification comes from the potential dependence of the process on rimer surface temperature rather than ambient temperature [Heymsfield and Mossop, 1985], but not to the extent that our initial rime splintering temperature weightings ($w_{RS}$) reached. We have amended $w_{RS}$ to the typical triangular weighting between -3 and -8°C and a uniform one between 0 and 10°C. The latter is still somewhat generous in order to pick up on any potential "cascade effect" between a droplet shattering or ice-ice collisional breakup "trigger" and rime splintering. We note this specifically in Section 2.3:

*We add a second, uniform temperature weighting (UNI) between 263 and 273 K to investigate the possibility of a droplet shattering or ice-ice collisional breakup `trigger' that feeds into a rime splintering `cascade'. The rimer surface temperature may in fact be the more important factor and can remain between 265 and 270 K, even for cloud temperatures a few degrees colder [Heymsfield 1984].*

c.  Also, there is no mention of the recent work on ice-ice collisions, and its parameterization by Phillips et al. (Phillips et al., JAS, 2017 and 2017) or for shattering of freezing drops ((Phillips et al., JAS, 2018). How do their parameterizations compare to those used here, and how might that influence differences in the effects upon precipitation?

Thank you for pointing out this oversight. As noted above (in response to point 3a), we have noted the existing breakup parameterizations in an expanded Section 2.2. To the end of this section, we note the differences in the importance of temperature, as our formulation depends directly and solely on temperature: *We expect a strong influence of temperature from our breakup tendency ($\delta N_{ice}/\delta t)_{BR}$ than was discussed in Phillips et al. 2017b, given the direct and sole dependence in Equation 5.*

4. It is stated that Crosier et al. noted fall streaks at cloud top in the radar measurements. This would seem to imply a seeding mechanism of ice from above that could also have fallen to the observation level of the aircraft, unless this has somehow been ruled out?

Thank you for this suggestion. Generally, an $N_{i,pri}$ peak at an altitude and an $N_{ice}$ peak at a slightly lower altitudinal band should be a signature of seeding. The new Figure 7 (old Figure 4) could address this somewhat, but to be more thorough we have included two supplemental figures that show the vertical profiles of $N_{ice}$ and $N_{i,pri}$ over time. For example, three locations near CFARR are shown in panels a, b, and c here with the darker colors corresponding to earlier times and lighter ones to later times:

[Figure]

*Figure S3 Temporal evolution of $N_{ice}$ (panels a, b, c) and $N_{i,pri}$ (panels d, e, f) profiles in the RS1 simulation from three, randomly-sampled latitude / longitude locations in the vicinty of CFARR. Eight profiles are shown for each location, one for each half hour from 18:00 UTC to 21:30 UTC with the darker colors representing earlier times and the lighter ones later times.*

Around cloud top, we do see that the peak in $N_{ice}$ extends below that in $N_{i,pri}$, but these temperatures are too cold for droplets to exist, so this should just be the signature of ice sedimentation. There are secondary peaks in both $N_{ice}$ and $N_{i,pri}$ at lower altitude, but both also exhibit a "pinch point" above this, i.e. there is not consistently a nucleation source about the low-level $N_{ice}$ peak. This low-level peak should mostly be due to secondary ice production.

We summarize this by stating the following in the new section on the Dynamic environment: *"these $Z_{DH}$ fall streaks, as well as those in differential reflectivity (shown in Crosier et al. 2014, their Fig. 5c) are signatures of ice crystal sedimentation and aggregation near cloud top. Ice crystal seeding may also be occurring with lower-level sedimentation, but the altitudinal peak in $N_{i,pri}$ does not fall consistently above that in $N_{ice}$ (Figs. S3 and S4) so that secondary ice production must generate a portion of this low-level ice."*

5. I would contend that most "larger-scale models" do not have two-moment microphysical schemes, so that the suggestion at lines 21-24 on page 10 are not practical.

Yes, you are right; within the CMIP5 models, the majority still use single-moment cloud microphysics schemes. But two-moment schemes generally lead to better representation of factors like greenhouse gas warming trends, snowfall intensity, and stratiform precipitation extent [Ekman 2014, Molthan and

Colle 2012, Morrison et al. 2009]. The IPCC has summarized its Fifth Assessment Report by saying that "climate models are incorporating more of the relevant aerosol-cloud interaction processes than at the time of AR4, but confidence in the representation of these processes remains weak." We take these results and trends to indicate that future model development will favor the incorporation of two-moment schemes. Indeed, given the uncertainties still associated with secondary ice production parameterizations, their model inclusion should generally come after the shift to a two-moment scheme.

Nevertheless we qualify our recommendation and note, as you mention above, that it may be more helpful initially to identify thermodynamic regimes under which these processes need to be included:

*As meso- and large-scale models transition toward two-moment cloud schemes, secondary ice production could be included in parameterizations with criteria for the number concentration of large hydrometeors: the droplet shattering and collisional breakup parameterizations are activated only for those cloudy grid cells with more than a threshold concentration of large droplets and graupel, hail, or snow respectively. In one-moment schemes, parameterizations on the basis of favorable thermodynamic regimes will be more useful for the time being.*

6. Comparison of observations and modeling: unless the authors can justify that the simulated movement of the rainband, and its dynamical nature, was very similar to that observed, it would make the comparisons shown in Fig. 4 not very useful. (If the comparison is not "fair", it could even be the case that the model IS producing sufficient secondary ice, for example, if the dynamical / thermodynamical conditions of the simulated clouds in that region and over that time are different than observed!) The description of the analysis for panels b and c on page 12 is helpful, but the reader cannot see what is being compared (which is why some comparison of the observed radar evolution to the simulated one is needed early on in the paper.) The temperature ranges also need to be specified in Fig. 4, too.

This is an important and difficult point that you raise again. We have provided a description of the simulated dynamics in our response to your first comment and see that the embedded updrafts, the preceding surface winds, and the reflectivities within the rainband are all weaker than in the observations. But it is also the case that the old Figure 4 (now Figure 6) intended to show how important the secondary production is relative to primary nucleation in the model. We were considering the simulated fields only because there is no way to make an analogous comparison with the available data. However, we still note in the rewritten discussion that "*underestimation of the updrafts within the rain band core (Sec. 4.1) will lead to errors that offset each other somewhat: too few raindrops will form when the vertical velocity and supersaturation are too low, but these will also be lofted more quickly through the altitudinal band where rime splintering is favorable, leaving less time for collisions to occur.*"

In general, this kind of "dynamical buffering" for secondary ice production (i.e. that they require large hydrometeors but that also sufficient time in an appropriate temperature zone) should give some "dynamical resilience" to the simulations. In discussion of the altitudinal profiles of $N_i$, we also point out that "*The underestimated updrafts and radar reflectivities noted above in Section 4.1 may also help explain the too low $N_i$ around 2 km: larger vertical velocities could loft graupel to high altitudes and boost the contribution from collisional breakup.*

To aid with visualization of what is shown in the new Figure 7 (altitudinal profile, histogram, and time series of $N_{ice}$), we have made the outline of the subdomain surrounding CFARR more prominent in Figures 5 and 6 and mentioned explicitly that we are drawing from this subdomain in the figure caption.

We have also switched Figure 6 to an altitudinal slice rather than a pressure level. We indicate in the caption now that "*The median temperature at this altitude is 255 K with a minimum value of 245 K and a maximum value of 267 K.*"

7. Ice production rates: it needs to be stated more clearly how these derived from both the observations and the modeling results. Right now, it is unconvincing that this comparison is valid. Also, it should be stated somewhere that the CIP-15 observations were corrected using an algorithm designed to remove shattering artifacts, but some still likely remain because anti-shattering tips were not used on the instrument, as stated by Crosier et al.

Yes, thank you for pointing out that the details of this calculation were missing. We had used a centered finite difference:

$$\left.\frac{dN_{ice}}{dt}\right|_{t_i} = \frac{N_{ice}\,(t_{i+1}) - N_i(t_{i-1})}{t_{i+1} - t_{i-1}}$$

But we agree that this analysis is not as informative as focusing on the temperature-dynamic regimes where ice production is largest. For that reason, we remove the ice production rate figure.

Within what is now Section 4.2 on Ice production rates, we also describe the CIP-15 measurements more thoroughly:

*IAT algorithms, with those particles below a threshold IAT of $10^{-4}$ s classified as artifacts, were used to correct the ICNCs [Field et al. 2006b]. No shatter-resistant tips were used on the probe, but given the strict IAT threshold, the possibility of artifacts is limited.*

8. Qualifiers / limitations: need to be stated clearly throughout the paper. For example:

a. The model uses the primary ice nucleation parameterization of Phillips et al. (2008). Since INP measurements were not collected, it is unknown if this is an accurate representation, or not, and this might greatly affect the ratios of secondary to primary nucleated ice, including the possible importance of secondary ice to precipitation.

Yes, this is a good point to include. In Section 3 on the Simulation setup, we add

*Previous studies have noted that limited nucleating efficiencies in the PDA08 may lead to underestimation of ICNC at mixed-phase conditions [Barahona et al. 2010, Curry and Khvorostyanov 2012, Morales-Betancourt et al. 2012]. No ice nucleating particle (INP) measurements were made during this case study, but from other observational datasets, PDA08 still yields better agreement with in-situ ICNCs than purely lab-based or theoretical parameterizations [Sullivan et al. 2015].*

Here it is also important to point out that Crosier et al. have noted the potential for large contributions of homogeneous nucleation in the convective regions. We point out right after the comment above that

*Crosier et al. also note that the low cloud top temperatures and strong updrafts in convective regions generate supersaturations that could favor large ice production from homogeneous nucleation. While not observationally confirmed, these conditions could buffer the ice nucleation tendency to our choice of parameterization.*

In what is now Section 4.2 on ice production, we also add the following caveat:
*The magnitude of these values is subject to uncertainty from the nucleation parameterization, which, as noted above, has underestimated INP numbers in previous studies.*

b. To show an appreciable effect on surface precipitation (20% increase), rime splintering had to be increase over that typically depicted in models based on the laboratory measurements (e.g. Cotton et al. 1986 parameterization).

We have only left one test with an extended rime splintering where rime splintering is permitted to happen for 2 K below the typical threshold of -8°C. Earlier we had also made the error of showing the precipitation accumulation at 18:00 UTC ("as the rainband begins to pass over the UK" stated in the caption). We now look at the accumulation from the final model output file (23:30 UTC) after complete rainband passage and see accumulation differences on the order of 10 mm. Precipitation intensity differences remain about ±5 mm h$^{-1}$ for a maximum simulated intensity of 30 mm h$^{-1}$ (the 20% you cite) for the adjusted simulations.

c. Reasons for why the other two secondary ice processes might be less important here: (i) minimal graupel, which is important for ice-ice collisional breakup; (ii) limited number of raindrops? (Not sure what else would have limited that process here, but it would be good to know.) Also, it should be explained that the Crosier et al. study noted limited graupel in their observations, and I don't think they found any evidence of shattered frozen raindrops, but they clearly state that the former could have been limited by the inability of the aircraft to fly in the more convective regions.

Thank you for your thoughts. Building off these, we have included the spatial fields of graupel and snow mixing ratios and rain drop number concentrations in a new supplemental Figure:

[Figure]

*Figure S5 Graupel mixing ratio (a), snow mixing ratio (b), large-scale graupel quantity (c), and rain drop number concentration (d) in the simulation domain at 18:00 UTC for the RS2 simulation.*

From these, we agree that low graupel and raindrop concentrations at the appropriate altitudes are limiting the $N_{i,sec}$ from collisional breakup and droplet freezing. To Section 4.2 we add the following:

*If graupel were present at higher altitudes, $N_{i,sec}$ from breakup could increase significantly, as both the snow mixing ratio and fragment number parameter increase at colder temperatures. Similarly very limited raindrop number concentrations exist at the altitude where shattering probability is non-negligible. This importance of large hydrometeor number concentration for $N_{i,sec}$ suggests a means of parameterizing secondary ice production as meso- and large-scale models transition toward two-moment cloud schemes: the droplet shattering and collisional breakup parameterizations could be activated only for those cloudy grid cells with more than a threshold concentration of large droplets and graupel, hail, or snow respectively. In one-moment schemes, parameterizations on the basis of favorable thermodynamic regimes will be more useful for the time being.*

d. The enhancement of the updrafts and precipitation, and downdrafts, mentioned on page 14 needs to be backed up with some evidence.

We have deleted this statement because we are not showing a feedback of the ice production on dynamics yet. With the updraft field in the new Figure 3 we can only state that the regions of highest precipitation intensity (and precipitation intensity deviation) occur in the regions of highest vertical velocity.

e. Some discussion of the model resolution effects should be included, for both properly simulating the microphysics as well as the dynamics.

Within the discussion section, we have added discussion on the potential impact of resolution and the particular microphysics scheme on our results:

*The choices of spatiotemporal resolution and microphysics scheme are particularly important for convective clouds because of the fine structure of precipitation and strong feedbacks between hydrometeor formation, latent heating, and cloud dynamics. Previous studies have generally found a spatial resolution of 4 to 6 km to be sufficient to reproduce precipitation extremes [e.g., Prein et al. 2013, Pieri et al. 2015]. This resolution dependence results from changes in the vertical moisture advection, in turn due to adjustments of vertical velocity with resolution [Yang et al. 2014]. For simulations whose resolutions border on the ``gray zone" scales (around a tenth of a degree), over-representation of convective activity is possible by both the parameterization and explicit resolution [Pieri 2015]; however, our simulations are at a fully convection-permitting scale and use only a reduced form of the Tiedtke mass-flux scheme for shallow convection [Tiedtke 1989] and this should not be a concern.*

*The use of a two-moment scheme is also important for simulation of extreme precipitation [Otkin 2006]. Certain one-moment schemes tend to generate overly large drops and too high precipitation rates [Thompson 2004], but SB06 tends to produce especially large quantities of graupel [Otkin et al. 2006]. The Bigg parameterization, as a precursor to our droplet shattering additions, has been shown in previous studies to predict very low numbers of frozen drops [e.g., Morrison et al. 2005, Fan et al. 2009], which may contribute to underestimate of secondary ice here. The more rigorous alternative would be to account for immersed surface area and scavenging of ice nucleating particles as in [Paukert and Hoose 2014, Paukert et al. 2017], and future work should implement both an updated immersion freezing and secondary ice parameterizations.*

Technical Corrections:

1. Table 2 and Fig 2 are confusing, since most of these runs are not discussed in the paper. I would suggest only showing those that are discussed here, and just noting somewhere (if important) that other variations did not show much change in the results. Also, if the text could explain the naming convention for the different simulations used, that would be much easier for the reader to interpret them.

In part because no results were shown from several simulations and in part because some parameter values were too extreme, we have adjusted the simulations. These changes are listed in a new Table 2:

**Rime splintering**

| | | |
|---|---|---|
| RS1: | $\aleph_{RS} = 300$, | $w_{RS} = \text{TR}$ |
| RS2: | $\aleph_{RS} = 300$, | $w_{RS} = \text{UNI}$ |

**Ice-ice collisional breakup**

| | |
|---|---|
| BR1ig: | *graupel_breakup_ice* |
| | $F_{BR} = 180$, $T_{min} = 256$, $\gamma = 3$ |
| BR2ig: | *graupel_breakup_ice* |
| | $F_{BR} = 360$, $T_{min} = 249$, $\gamma = 5$ |
| BR2sg: | *graupel_breakup_snow* |
| | $F_{BR} = 360$, $T_{min} = 249$, $\gamma = 5$ |

**Droplet shattering**

| | | | |
|---|---|---|---|
| DS1: | $\aleph_{DS} = 2$, | $p_{max} = 5\%$, | $\sigma = 3$ |
| DS2: | $\aleph_{DS} = 10$, | $p_{max} = 10\%$, | $\sigma = 5$ |

**Combinations**

| | |
|---|---|
| RS2 + BR2ig: | $\aleph_{RS} = 300$, $w_{RS} = \text{UNI}$ |
| | *graupel_breakup_ice* |
| | $F_{BR} = 360$, $T_{min} = 249$, $\gamma = 5$ |
| DS2 + BR2ig: | $\aleph_{DS} = 10$, $p_{max} = 10\%$, $\sigma = 5$ |
| | *graupel_breakup_ice* |
| | $F_{BR} = 360$, $T_{min} = 249$, $\gamma = 5$ |
| RS2 + DS2: | $\aleph_{RS} = 300$, $w_{RS} = \text{UNI}$ |
| | $\aleph_{DS} = 10$, $p_{max} = 10\%$, $\sigma = 5$ |

**Control**

| | |
|---|---|
| CTRL: | $\aleph_* = 0$, $F_{BR} = 0$ |
| ALL: | $\aleph_{RS} = 300$, $w_{RS} = \text{UNI}$ |
| | $F_{BR} = 360$, $T_{min} = 249$, $\gamma = 5$ |
| | *graupel_breakup_\** |
| | $\aleph_{DS} = 10$, $p_{max} = 10\%$, $\sigma = 7$ |

The previous naming schemes were based on the idea of four levels of experiments – **C**onservative, **M**oderate, **N**arrow, and **G**enerous – but this was not clear since there was no description in the manuscript. The idea had been to have a fractional factorial experimental design that probed the whole parameter space without an expensive, one-at-a-time approach. In the new scheme shown above, we have only defined two "levels" for each process to avoid the most extreme values we had used earlier, particularly for rime splintering and droplet shattering.

We have also condensed Figure 2 for the new simulation layout:

[Figure]

*Figure 2 Fragment numbers, weightings, and probabilities from the secondary ice production parameterizations. In panel a, we show $N_{BR}$ from both ice-ice collisional breakup simulations (BR1 and BR2) as well as the triangular and uniform $w_{RS}(T)$. In panel b, we show $p_{DS}$ from both droplet shattering simulations (DS1 and DS2) and $w_{RS}$ once again. The overlapping temperature regions of these are particularly important to understand any feedback between the processes.*

If the major issues in this review are addressed, I suspect that much of the wording will be revised, and thus I will refrain from noting any suggestions at this time.